# P-LSHv2: a multi-decadal global daily land surface actual evapotranspiration dataset enhanced with explicit soil moisture constraints in remote sensing retrieval

Jin Feng[1,3], Ke Zhang[1,2,3,4*], Lijun Chao[1,3], Huijie Zhan[1], Yunping Li[1,3,5]

[1] National Key Laboratory of Water Disaster Prevention, and College of Hydrology and Water Resources, Hohai University, Nanjing, Jiangsu, 210024, China

[2] Yangtze Institute for Conservation and Development, Hohai University, Nanjing, Jiangsu, 210024, China

[3] China Meteorological Administration Hydro-Meteorology Key Laboratory, Hohai University, Nanjing, Jiangsu, 210024, China

[4] Key Laboratory of Water Big Data Technology of Ministry of Water Resources, Hohai University, Nanjing, Jiangsu, 210024, China

[5] College of Water Resources, North China University of Water Resources and Electric Power, Zhengzhou 450046, China

*Correspondence to*: Ke Zhang (kzhang@hhu.edu.cn)

**Abstract.** Accurately quantifying the impact of soil water availability on evapotranspiration (ET) is curcial for improving ET retrieval accuracy. However, most global satellite-derived ET datasets do not explicitly incorporate soil moisture constraints, leading to significant uncertainties, particularly in water-limited regions. In this study, we propose an enhanced soil moisture constraint scheme that effectively captures soil moisture's influence on vegetation transpiration and soil evaporation using a quantile-based approach. Unlike previous methods, this scheme relies solely on soil moisture data, reducing uncertainties associated with heterogeneous soil hydraulic properties. We integrated this approach into the process-based land surface ET/heat fluxes algorithm (P-LSH, or P-LSHv1), developing an improved version, P-LSHv2. Using observations from 106 global flux towers, we calibrated biome- and climate-specific parameters and quantified moisture constraints across diverse climates and land cover types. P-LSHv2 achieves notable improvements in ET estimation, with a reduced Root Mean Square Error (RMSE) of 0.67 mm d$^{-1}$ and an increased correlation coefficient (R) of 0.81, outperforming its predecessor, P-LSHv1, particularly in arid regions. Comparative analyses show that P-LSHv2 surpasses the Penman-Monteith-Leuning model and the Global Land Evaporation Amsterdam Model in capturing soil moisture anomalies' effects on ET, enhancing global ET accuracy. Leveraging the P-LSHv2 algorithm, we have produced a long-term global daily ET dataset spanning 1982–2023, providing a valuable resource for research on terrestrial water and energy cycles and climate change. The dataset is freely available at https://doi.org/10.11888/Terre.tpdc.301969 (Feng Jin, 2025).

## 1 Introduction

Evapotranspiration (ET) is considered the second largest component of the terrestrial water cycle after precipitation and also plays a critical role in atmospheric-terrestrial carbon and energy exchanges (Jung et al., 2010; Wang and Dickinson, 2012;



Zhang et al., 2016). Therefore, accurate estimation of global ET is essential for improving our understanding of the surface energy budget, global water cycle, and climate change (Fisher et al., 2017; Oki and Kanae, 2006).

Among various ET estimation methods, remote sensing retrieval provides an effective means to estimate large-scale ET for its
spatially continuous and temporally frequent measurements of surface biophysical variables (Fisher et al., 2008; Longo et al., 2019). Several remote sensing-based global ET datasets have been generated across different temporal spans, spatial scales and resolutions (Jung et al., 2011; Martens et al., 2017; Mu et al., 2011; Zhang et al., 2010; Zhang et al., 2019). Microwave-based measurements of hydrology states inside the canopy or soil generally contribute to high temporal resolution (i.e., daily) of ET estimation but typically have coarse spatial resolution (Martens et al., 2017; Wang et al., 2019), which is unsuitable for
evaluating crop water requirements over complex agricultural landscapes. Conversely, the Moderate Resolution Imaging Spectroradiometer (MODIS), which provides a high spatial resolution of vegetation observation (i.e., 500 m or finer), offers finer ET estimates (Mu et al., 2011; Zhang et al., 2019). However, MODIS data do not cover the pre-2002 period and are insufficient for long-term interannual variability and attribution analysis of ET. Therefore, before MODIS-era, how to extend high-quality ET series in the case of Advanced Very High-Resolution Radiometer (AVHRR), and how to balance fine spatial
and high temporal resolutions, are essential for extending the coverage of ET datasets.

An additional challenge lies in the reliability and robustness of ET datasets, particularly in complex global ecosystems and diverse climates. Although some global ET datasets have been developed, stark discrepancies are identified among various datasets in terms of magnitude and changing trends (Badgley et al., 2015; Li et al., 2022; Pan et al., 2020; Zhu et al., 2022). For instance, an investigation from the Global Soil Moisture Project-2 (GSWP-2) found that the global average annual ET
from 15 models varied widely, ranging from 272 to 441 mm yr$^{-1}$, with the maximum value 1.5 times the minimum (Dirmeyer et al., 2006). Similarly, Jiménez et al. (2011) showed considerable disagreement among ET estimates from 12 global products, particularly in tropical rainforests. McCabe et al. (2016) highlighted that four widely-used remote sensing ET models exhibit significant variability across specific biomes and climate zones, with no model consistently outperforming the others. The Water Cycle Multi-Mission Observation Strategy-Evapotranspiration Project (WACMOS-ET) demonstrated low correlation
and disagreement between various products in arid regions (Michel et al., 2016), and the MODIS product tends to underestimate tropical and subtropical ET fluxes, with obvious disagreement among various products in water-limited situations (Miralles et al., 2016). Chen et al. (2014) further found that the annual average ET across eight products in China ranged from 535 to 852 mm yr$^{-1}$, with higher uncertainties in tropical and humid regions. In summary, despite the numerous remote sensing ET models and datasets currently available, each has its own merits and disadvantages, and no consensus on
which method is the best has been reached.

In previous remote sensing ET algorithms, the vapor pressure deficit was generally adopted to reflect the associated moisture constraint on canopy conductance (Mu et al., 2011; Zhang et al., 2009; Zhang et al., 2010; Zhang et al., 2019). However, several studies have shown that water availability is the primary limiting factor for ET in arid areas (Yang et al., 2019; Zhang et al., 2015), namely the so-called water-limited regions, where soil moisture constraints on canopy conductance increase with
climate dryness and drought severity(Han et al., 2020; Novick et al., 2016). Due to the water potential gradient between leaf





and air, water is transported from soil to vegetation roots, and leaves, and then dissipated into the atmosphere through stomata. Therefore, soil water content serves as the direct water pool for vegetation and regulates the magnitude of water extracted by vegetation roots (Feng et al., 2022; Liu et al., 2020b). Although vapor pressure deficit and soil moisture are connected through land-atmosphere interactions (Liu et al., 2020a; Zhou et al., 2019), their anomalies may be temporally lagged (Zhang et al.,
2022b). Therefore, incorporating soil moisture constraints is essential for improving ET estimation at finer temporal scales (Fu et al., 2022a). Additionally, this connection may decouple under global warming and climate change scenarios(Liu et al., 2020b), further underscoring the significance of soil moisture constraints in ET estimation.

One of the main challenges of incorporating soil moisture constraints into remote sensing ET algorithms is the development of a robust and globally applicable moisture constraint scheme for quantifying water availability. Although soil moisture
constraints have been applied to estimate ET at local (Xu et al., 2018; Zhu et al., 2014), regional, and global scales (Brust et al., 2021; Purdy et al., 2018), long-term available datasets are limited by the lack of universal constraint schemes and soil moisture datasets. Furthermore, existing soil moisture constraint schemes generally rely on soil hydraulic properties as thresholds (Martens et al., 2017; Zheng et al., 2022). However, global soil data present significant uncertainties, which substantially affect ET estimation (Dennis and Berbery, 2021). On the one hand, gridded datasets derived from field
investigation are prone to great uncertainties, particularly in high-altitude and other harsh regions (Dai et al., 2019). On the other hand, the pedotransfer functions for generating gridded soil properties still lack proper extrapolation methods from local to global scales (Ma et al., 2021; Van Looy et al., 2017).

In response to these challenges, we integrated a new soil moisture constraint scheme into the previously developed Process-based Land Surface evapotranspiration/Heat fluxes algorithm (P-LSH, or P-LSHv1) (Zhang et al., 2010; Zhang et al., 2015),
creating the second version (P-LSHv2). The objectives of this study are to (1) develop a globally universal soil moisture constraint scheme, calibrate key parameters, and quantify moisture levels across diverse vegetation types and climate zones, (2) upgrade ET algorithm by incorporating soil moisture constraints and evaluate its robustness from sites to basins, and (3) generate a long-term, reliable global daily ET dataset spanning from 1982 to 2023.

## 2 Methodology

### 2.1 Baseline description of the P-LSH algorithm

The P-LSH algorithm separately estimates three components of the ground surface evapotranspiration: vegetation transpiration ($\lambda E_{Canopy}$: W m$^{-2}$), soil evaporation ($\lambda E_{Soil}$: W m$^{-2}$), and open water evaporation ($\lambda E_{Water}$: W m$^{-2}$). These components are calculated based on the pixel's land cover type. In vegetation pixels, the ET is partitioned into vegetation transpiration and soil evaporation by partitioning available energy according to the fractional vegetation cover. In barren and open water pixels, ET
consists solely of soil evaporation and open water evaporation, respectively.

The Penman-Monteith equation is used to calculate vegetation transpiration:



$$\lambda E_{\text{Canopy}} = \frac{\Delta A_c + \rho C_p VPD g_{ac}}{\Delta + \gamma(1 + g_{ac}/g_c)}, \tag{1}$$

where $\lambda$ (J kg$^{-1}$) is the latent heat of vaporization, $\Delta$ (Pa K$^{-1}$) is the slope of the saturated water vapor pressure curve as a function of temperature; $A_c$ (W m$^{-2}$) is the available energy allocated to the canopy; $\rho$ (kg m$^{-3}$) is air density; $C_p$ (J kg$^{-1}$ K$^{-1}$) is

the specific heat capacity of air; VPD (Pa) is vapor pressure deficit; $g_{ac}$ (m s$^{-1}$) is the aerodynamic conductance of the canopy; $\gamma$ (-) is the psychrometric constant; $g_c$ (m s$^{-1}$) is the canopy conductance calculated using the Jarvis-Stewart-type model:

$$g_c = g_0(NDVI) \times m(T_{\text{day}}) \times m(VPD) \times m(C_{CO_2}), \tag{2}$$

$$g_0(NDVI) = 1/[b_1 + b_2 \times exp(-b_3 \times NDVI)] - 1/(b_1 + b_2), \tag{3}$$

where $g_0$ (m s$^{-1}$) is the maximum value of $g_c$ based on NDVI; $m(T_{\text{day}})$, $m(VPD)$, and $m(C_{CO_2})$ are stress factors associated with

daylight temperature $T_{\text{day}}$ (°C), VPD (Pa), and $CO_2$ concentration (ppm), respectively; $b_1$, $b_2$, and $b_3$ are biome-and-climate-specific parameters derived from flux towers measurements. In addition, $T_{\text{opt}}$ (°C) and $\beta$ (°C) in $m(T_{\text{day}})$ calculation represent optimal air temperature for photosynthesis and empirical parameters, respectively, which are taken as the biome-and-climate-specific parameters in this study. More details of stress factors are available at Zhang et al. (2010) and Feng et al. (2022). The soil evaporation is reduced from its potential value using the moisture constraint:

$$\lambda E_{\text{Soil}} = f\lambda E_{POT}, \tag{4}$$

$$f = RH^{(\frac{VPD}{k})}, \tag{5}$$

$$\lambda E_{POT} = \frac{\Delta A_s + \rho C_p VPD g_{as}}{\Delta + \gamma \times g_{as}/g_{\text{totc}}}, \tag{6}$$

where $f(-)$ is the moisture constraint estimated by the complementary relationship hypothesis here; RH (-) is the relative humidity; k (Pa) is a parameter to fit the complementary relationship and reflect the sensitivity of VPD, we regard it as a

parameter for different land cover types and climates. The $\lambda E_{POT}$ (W m$^{-2}$) is the potential soil evaporation; $A_s$ (W m$^{-2}$) is the available energy component allocated to the soil; $g_{as}$ (m s$^{-1}$) is the aerodynamic conductance of the soil surface and is defined as the sum of the conductance associated with convective heat transfer ($g_{ch}$: m s$^{-1}$) and the conductance to radiative heat transfer ($g_{rh}$: m s$^{-1}$). The $g_{ch}$ term is expressed in its resistance form $r_c$, a biome-and-climate-specific parameter, while the $g_{rh}$ term is calculated by daylight temperature. The $g_{\text{totc}}$ (m s$^{-1}$) is the corrected value of total aerodynamic conductance $g_{\text{tot}}$ (m s$^{-1}$) and the

correction is based on actual air temperature and pressure. In this study, the $g_{\text{tot}}$ term is also expressed in the form of total aerodynamic resistance $r_{\text{tot}}$, which is regarded as a parameter determined by land cover types and climates.

The open water evaporation is calculated using a modified Penman equation that considers the effects of surface wind speed on aerodynamic conductance:

$$\lambda E_{\text{Water}} = \frac{\Delta A + \rho C_p VPD g_{aw}}{\Delta + \gamma}, \tag{7}$$



where A (W m$^{-2}$) is the available energy for open water; $g_{aw}$ (m s$^{-1}$) is the aerodynamic conductance of the open water estimated by wind speed:

$$g_{aw} = \frac{1+0.536U_2}{4.72[ln(z_m/z_0)]^2},$$
(8)

where $U_2$ (m s$^{-1}$) is the wind speed at 2 m height converted from measurement height $z_m$ (m) using vertical wind speed function; $z_0$ (m) is the aerodynamic roughness of the water surface and set to 0.00137. Further details of the P-LSH algorithm are

available in Zhang et al. (2010) and Feng et al. (2022).

**2.2 Improvements on the P-LSH algorithm**

In this study, a series of improvements are implemented to evolve P-LSHv1 into the P-LSHv2 algorithm, incorporating soil moisture constraints on both vegetation transpiration and soil evaporation, as well as a Bayesian and Sobol' uncertainty analysis framework. These improvements allow for a more explicit quantification of water supply limitations on ET and offer

a structured approach to parameter uncertainty across diverse land covers and climates.

**(1)    Soil moisture constraint scheme for vegetation transpiration**

Previous studies (Liu et al., 2020b) have revealed that atmospheric moisture and soil moisture are generally decoupled in arid areas, which introduces significant uncertainty in quantifying soil water supply on ET. In this study, vegetation transpiration response to soil moisture is represented by the typical Jarvis-Stewart model (Jarvis, 1976; Stewart, 1988), in which a revised

soil moisture constraint scheme is applied to constrain canopy conductance:

$$g_c = g_0(NDVI) \times m(T_{day}) \times m(VPD) \times m(C_{CO_2}) \times m(SM),$$
(9)

$$m(SM) = \begin{cases} \frac{SM-SM_{min}}{SM_c-SM_{min}} & SM \leq SM_c \\ 1 & SM > SM_c \end{cases},$$
(10)

$$SM_c = F(SM, n),$$
(11)

where m(SM) is the soil moisture stress factor, calculated by actual surface soil moisture (SM: m$^3$ m$^{-3}$), its critical value $SM_c$

(m$^3$ m$^{-3}$), and its minimum value $SM_{min}$ (m$^3$ m$^{-3}$) over a given period. Such a linear equation ensures the record of soil moisture dynamic characteristics as well as the simple normalization structure.

Different from other studies, the $SM_c$ is calculated using the SM sequence, the quantile ($n$), and the quantile function ($F$) in this study, shown in Eq. (11). The $SM_{min}$ and $SM_c$ derived from quantiles and actual sequence are used as the threshold, which circumvents the use of soil hydraulic parameters that exhibit significant uncertainty. The parameter $n$ represents the proportion

of the period during which ET is subject to moisture constraints. Specifically, 100 indicates that ET is constantly constrained by soil moisture, while 0 indicates that ET is never constrained by soil moisture. Considering the potential impacts of different vegetation types and climates on constraints, we regard $n$ as a biome-and-climate-specific parameter, which is first calibrated by the tower measurements, and subsequently applied to global ET estimation. That means, the quantile $n$ takes the same value



for specific land cover types and climates, but when mapped to various SM sequences in global grid estimation, the

corresponding $SM_c$ may vary. In addition to quantiles, this scheme also relies on cumulative distribution function of the SM

sequence. The actual SM sequences used in this scheme are sufficient enough so that several heavy rainfall will not lead to

significant changes in distribution function unless the local climate changes significantly. Even so, the variations in $SM_c$

resulting from changes in the distribution function under climate change also reflect how our scheme responds to moisture

constraints in a changing climate. This scheme effectively quantifies the moisture constraint level of the land cover types and

climates, while also mitigating the uncertainty associated with soil hydraulic parameterization. The soil moisture stress

function is illustrated in Fig. 1.

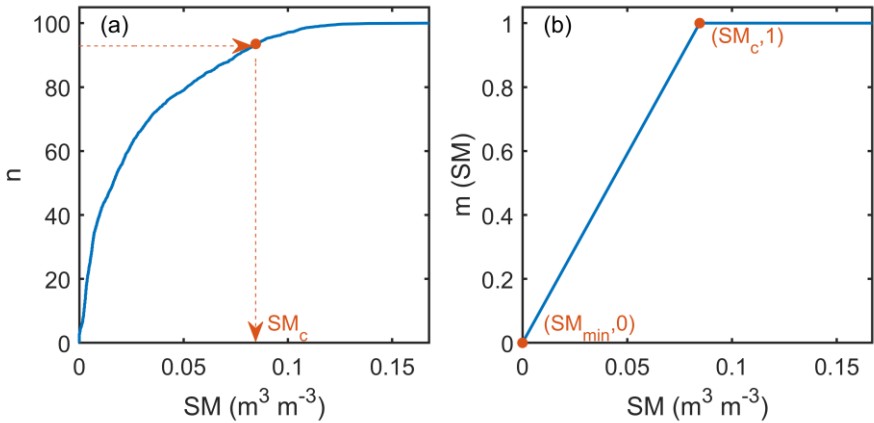

**Figure 1: Example illustration of (a) $SM_c$ estimation and (b) soil moisture stress function estimation.**

**(2)     Soil moisture constraint scheme for soil evaporation**

For soil evaporation, the original algorithm uses atmospheric humidity as a proxy for soil moisture constraint, which introduces

enormous uncertainty in estimating soil evaporation. In this study, a direct soil moisture constraint scheme following Feng et

al. (2023) is implemented to replace the moisture constraint in Eq. (4) for soil evaporation:

$$f = \frac{SM - SM_{\min}}{SM_{\max} - SM_{\min}}, \tag{12}$$

where $SM_{\max}$ (m³ m⁻³) and $SM_{\min}$ (m³ m⁻³) terms are the respective maximum and minimum SM sequences during a given

period. Eq. (12) is a specific version of Eq. (10) where the quantile $n$ reaches its maximum of 100 and $SM_c$ reaches $SM_{\max}$.

Here $f$ is a normalized soil moisture index (or saturation fraction), assuming that the recording period of soil moisture

observations can entirely cover the full range of soil moisture change at a given tower or pixel, which is suitably beneficial for

long-term decades of ET estimation.



**2.3 Parameters uncertainty analysis framework for the P-LSHv2 algorithm**

With the structural upgrades to the ET algorithm, it is necessary to perform a parameter uncertainty analysis and optimization in the P-LSHv2 algorithm. The Bayesian and Sobol' uncertainty analysis framework is employed to quantify parameter characteristics.

Sobol' sensitivity analysis is based on variance decomposition, which can quantify both the single impacts of parameters on the model outputs and the interaction impacts among multiple parameters. The variance of model output is based on variance

decomposition:

$$D(Y) = \sum_{i=1}^{d} D_i + \sum_{i<j}^{d} D_{ij} + \cdots + D_{12\ldots d}, \tag{13}$$

where D(Y) is the variance of model output Y; d is the total number of parameters; $D_i$ represents the partial variance for the first-order sensitivity of the $i^{th}$ parameter to the model output; $D_{ij}$ denotes the partial variance for the second-order sensitivity corresponding to the interaction between the $i^{th}$ and $j^{th}$ parameters; $D_{12\cdots d}$ indicates the partial variance for the $d^{th}$-order

sensitivity involving all parameter interactions.

The first-order index and the total-order index can reflect the impacts of single parameters and multiple parameters' interaction on the model output:

$$S_i = \frac{D_i}{D}, \tag{14}$$

$$S_{Ti} = 1 - \frac{D_{\sim i}}{D}, \tag{15}$$

where $S_i$ is the first-order index and represents the sensitivity from the effect of parameter $X_i$; $S_i$ is the total-order index and represents the sensitivity from the effect of parameter $X_i$ and its interactions with the remaining parameters; $D_{\sim i}$ is the amount of variance attributed to all other parameters after removing $X_i$.

After identifying sensitive parameters of the P-LSHv2 algorithm, it is necessary to quantify the uncertainty of the sensitive parameters. In recent years, the Bayesian theory has been widely used in parameter uncertainty analysis. Based on subjective

prior knowledge and samples, the prior distribution of parameters can be updated to the posterior distribution, which significantly reduces the uncertainty of prior ranges:

$$\pi(X|x) \propto f(x|X)\pi(X), \tag{16}$$

where x is the measurement sample; X is a group of parameter set; $\pi(X)$ is the prior distribution; $f(x|X)$ is the likelihood function.

Among various Bayesian-based methods, the Differential Evolution Markov Chain (DE-MC) method stands out due to its effective combination of the Differential Evolution algorithm and the Markov Chain Monte Carlo method. The DE-MC



algorithm performs well in terms of convergence speed and computational efficiency. In the DE-MC algorithm, candidate proposals are initially generated using two random chains, and their difference is scaled and added to the current chain:

$$X_p = X_i + \gamma(X_{R1} - X_{R2}) + \varepsilon, \tag{16}$$

where $X_p$ is the proposed parameter set; $X_i$ is the current parameter set; $X_{R1}$ and $X_{R2}$ are two random chains excluding $X_i$; $\gamma$ is the scaling factor; $\varepsilon$ is a term that reflects probability acceptance rules.

**2.4 Parameters setting and multi-scale evaluation of the P-LSHv2 algorithm**

Compared to P-LSHv1, the P-LSHv2 algorithm has been upgraded both in ET algorithm structure and biome-specific parameters. Considering the various responses of vegetation physiological processes, particularly with respect to soil moisture
constraints under different climates, the P-LSHv2 algorithm takes distinct parameters for dry and wet zones within each land cover type. While the algorithm structure remains consistent, specific parameters are adjusted based on diverse zones.

To evaluate the P-LSHv2 algorithm and demonstrate its advances relative to the P-LSHv1 algorithm, a multi-scale evaluation procedure is developed. First, we quantify parameter uncertainty and calibrate optimal values based on global 106 flux towers containing daily ET measurements (Sect. 4.1). The performance of P-LSHv2 algorithm is then evaluated through both direct
validation and leave-one-out cross-validation methods at 106 flux towers (Sect. 4.2). Second, we evaluate the P-LSHv2 algorithm at basins across the conterminous United States (CONUS) and other global regions, using the reconstructed values from the water balance method as the benchmark. The evaluation of basin scales is conducted on monthly, annual, and multiyear levels (Sect. 4.3). Finally, we generate global ET datasets using the P-LSHv2 algorithm and then compare them with those from P-LSHv1 algorithm, analyzing their diversity in spatial patterns and temporal trends (Sect 4.4).

**3 Data and materials**

**3.1 Eddy Covariance Flux Towers**

In this study, climatic and flux measurements from eddy covariance towers that represent nine different land cover types, as well as their associated dry or wet climate zones, are used to calibrate and verify algorithm performance. These land cover types include evergreen needleleaf forest (ENF), evergreen broadleaf forest (EBF), deciduous broadleaf forest (DBF), mixed
forest (MF), open shrublands (OSH), woody savannas (WSA), savannas (SAV), grasslands (GRA), and croplands (CRO). The majority of towers come from the FLUXNET2015 datasets (Pastorello et al., 2020), with additional towers from the AmeriFlux network (https://ameriflux.lbl.gov/) and the European Eddy Fluxes Database Cluster (European Flux) (http://www.europe-fluxdata.eu/). To ensure robustness, we applied the following selection criteria: (a) each land cover type and climate zone should contain at least two towers, and two-year or longer measurements are available for each tower; (b) the land cover type
from tower footprints should be consistent with the dominant type from coarse resolution remote sensing data; (c) the towers





should have both global integrity of biome and climate zone and regional representation. As a result, 106 flux towers are selected for the ET algorithm development and evaluation (Fig. 2). More details involving flux towers are available in Table A1 of Appendix A.

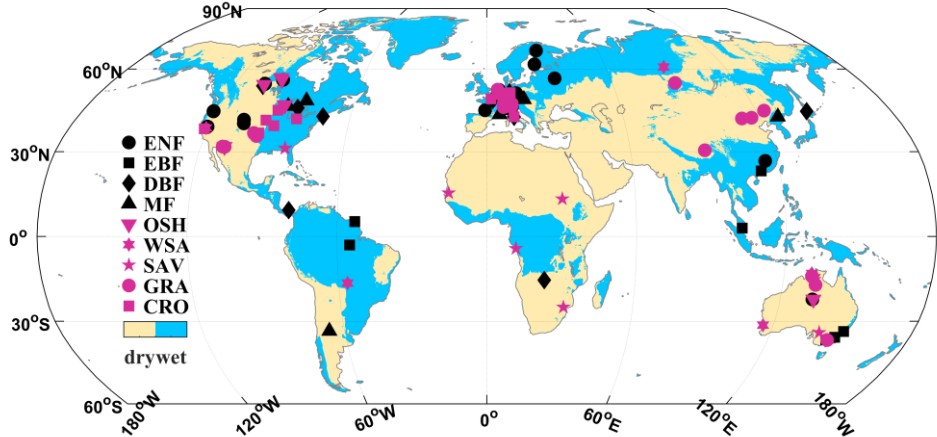

**Figure 2: Locations of 106 flux towers used for ET algorithm development and validation. The symbols represent land cover types and colored backgrounds represent climate zones.**

The tower measurements including air temperature, VPD, wind speed, net radiation, air pressure, surface soil moisture, and latent flux are used to drive and evaluate ET algorithm. The 30-minute measurements from AmeriFlux and European Flux are aggregated into daily averages when at least 80% of the measurements are available.

## 3.2 Climatic and remote sensing forcing data

The global datasets used for ET generation include the satellite remote sensing inputs, daily surface meteorology, and climate zone database. The satellite inputs consist of NDVI, net radiation, and land cover, while daily surface meteorology and soil moisture data come from reanalysis datasets.

The arid index (AI) data come from the Global Aridity Index and Potential Evapotranspiration Database - Version 3 (Zomer et al., 2022). Based on the AI and the distribution of flux towers, the globe is divided into dry (AI < 0.65) and wet zones (AI ≥ 0.65). This separation acknowledges that soil moisture constraints and parameters differ substantially between dry and wet zones, even within the same biome.

The NDVI sequence is merged from three independent products of the AVHRR GIMMS NDVI (Tucker et al., 2005), University of Arizona Vegetation Index and Phenology Lab (VIP) NDVI (Didan, 2010), and MODIS NDVI (Didan, 2015) using the equidistant cumulative distribution function (EDCDF) matching method (Zhang et al., 2015). The adjusted GIMMS and VIP sequences are fused with the continuously updated MODIS sequence to produce a consistent long-term 1/12° NDVI series. The temporal linear interpolation is applied to generate daily NDVI sequence from the original adjacent semi-monthly values, providing a simple but effective method to produce daily series of vegetation observations (Zhang et al., 2010).



The daily satellite-based net radiation term is derived from the NASA World Climate Research Programme/Global Energy
and Water Cycle Experiment (WCRP/GEWEX) Surface Radiation Budget (SRB) Release−3.0 datasets and the Clouds and the
Earth's Radiant Energy System (CERES) SYN1deg radiative fluxes. Since they are two independent datasets and neither
dataset spans the entire period since 1982, the same merging method used for NDVI is applied to fuse SRB with the latest
CERES. The 1° coarse resolution radiation inputs are further resampled to 1/12° resolution using the widely-used bilinear
interpolation method.

For land cover types, the MODIS Land Cover Dynamics product at 0.05° resolution (Sulla-Menashe et al., 2019) is selected.
To match the spatial resolution of other inputs, the land cover type is resampled to 1/12° resolution based on the dominant type
within each pixel.

The daily meteorological data, including air temperature, vapor pressure, and wind speed, are obtained from the NCEP/DOE
AMIP-II Reanalysis (NCEP2) (Kanamitsu et al., 2002). Despite its coarse spatial resolution of 1.9°×1.875°, NCEP2 is chosen
for its long-term consistency and generally high quality against ground observations (Zhang et al., 2015). The coarse-resolution
meteorology is further resampled to 1/12° resolution using bilinear interpolation. In addition, monthly $CO_2$ concentration
records come from The Global Monitoring Laboratory of the National Oceanic and Atmospheric Administration
(ftp://aftp.cmdl.noaa.gov/products/trends/co2/co2_mm_gl.txt).

The surface soil moisture data are derived from Global Land Data Assimilation System (GLDAS) Noah Land Surface Model
(Rodell et al., 2004) for its long-term temporal consistency and global spatial coverage. To cover the entire period from 1982
to 2023, we use two versions: version 2.0 for data prior to 2000 and version 2.1 for data from 2000 onwards. For version 2.1,
3-hourly soil moisture data are averaged to produce daily values.

### 3.3 Other remote sensing evapotranspiration and associated datasets

To evaluate and compare our ET estimates with other products, two mainstream remote sensing ET datasets with high
reputations are selected. The first is the Global Land Evaporation Amsterdam Model (GLEAM) Version 3.3a (Martens et al.,
2017), which is a microwave-based dataset with a 0.25° daily resolution, covering a long time span since 1980. The second
is the Penman-Monteith-Leuning (PML) Version 2 (Zhang et al., 2019), a diagnostic biophysical model that couples
transpiration with gross primary productivity and offers a 500 m resolution and an 8-day interval since 2002.

Our new ET estimates are also validated against the reconstructed ET estimates using the terrestrial water balance method at
the basin level. One dataset generated by Wan et al. (2015) provides monthly ET benchmarks of 592 subbasins across the
conterminous United States (CONUS). The ET item is regarded as the residual of precipitation, runoff, and downscaled water
storage changes. These subbasins, which range from 292 to 303,700 km², cover 73% of the CONUS, making the dataset well-
suited for evaluating remote sensing ET products. For our evaluation, we use the monthly sequence from 2003 to 2008 to
evaluate ET estimations from P-LSHv2, P-LSHv1, GLEAM, and PML, as well as other remote sensing ET datasets presented
in our previous study (Chao et al., 2021). Another dataset (Pan et al., 2012) provides monthly water budget estimates for 32
major global basins by merging numerous global datasets. Here we use its ET estimates, in which the imbalance error has been





resolved by the constrained Kalman filter technique. The 32 basins cover all continents except Antarctica and represent a wide range of climates and land covers, spanning from 1984 to 2006.

## 4 Results

### 4.1 Uncertainty analysis and optimization of parameters


We used ET measurements from 106 global flux towers to conduct an uncertainty analysis and optimize the parameters of the P-LSHv2 algorithm. Details about these towers are provided in Appendix A. The 106 flux towers cover 9 main land cover types, which represent 97.1% of the global land area, excluding permanent ice, snow, barren lands, and open water bodies. Each land cover type is further divided into dry and wet zones, where key parameters are identified and calibrated separately.

Since the dry zone constitutes only 1.7% of the EBF type, no further classification was made for EBF and there are 17 scenarios in total.

A Sobol' sensitivity analysis revealed significant variations in parameter sensitivity across multiple scenarios, as shown in Fig. 3. In general, parameters $b_1$ and $b_3$, which govern canopy conductance, demonstrate the highest sensitivity in the P-LSHv2 algorithm. The first-order indices for $b_1$ and $b_3$ are 26% and 18%, respectively, while their total-order indices are 55% and

40%. Next in sensitivity are $\beta$ and $T_{opt}$, both of which are associated with temperature stress factors in canopy conductance estimation, with total-order indices of 18% and 15%, respectively. Other parameters such as $r_{tot}$, k, n, $b_2$, and $r_c$, which are involved in soil evaporation and canopy conductance, exhibit relatively low sensitivity, with total-order indices of 8%, 5%, 5%, 2%, and 1% respectively. Parameters related to temperature and VPD thresholds ($T_{close\_min}$, $T_{close\_max}$, $VPD_{open}$, and $VPD_{close}$) are insensitive, with first-order and total-order indices close to 0. To summarize, among 13 parameters in the P-LSHv2

algorithm, $b_1$ and $b_3$ are the most sensitive, while $\beta$ and $T_{opt}$ have moderate sensitivity. The $r_{tot}$, k, n, $b_2$, and $r_c$ exhibit weak sensitivity, and $T_{close\_min}$, $T_{close\_max}$, $VPD_{open}$, and $VPD_{close}$ are found to be insensitive. Consequently, in subsequent parameter uncertainty analysis, these four insensitive parameters are removed, and the remaining nine sensitive parameters are retained for optimization.



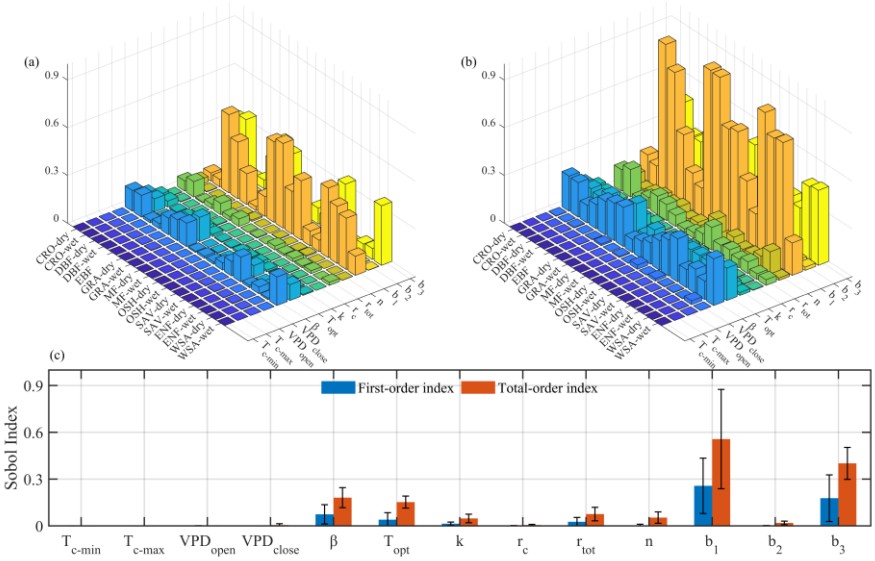

**Figure 3: The (a) first-order sensitivity index and (b) total-order sensitivity index of the parameters across diverse land cover types and climate zones. The average values of diverse zones are shown in (c).**

The Differential Evolution Markov Chain (DE-MC) method is used for parameter calibration, with the root mean square error (RMSE) as the objective function, as well as the Nash-Sutcliffe efficiency coefficient (NSE) and coefficient of determination ($R^2$) to quantify mismatches. 10 Markov chains were set to perform 20,000 iterations for parallel crossover operations, with a burn-in period of 5,000 iterations. Each parameter can converge stably across all 17 scenarios. The posterior distributions of parameters, with 95% high probability intervals, are shown in Fig. 4. Compared with the wider prior intervals, most parameters show significant updating and reduction in uncertainty.

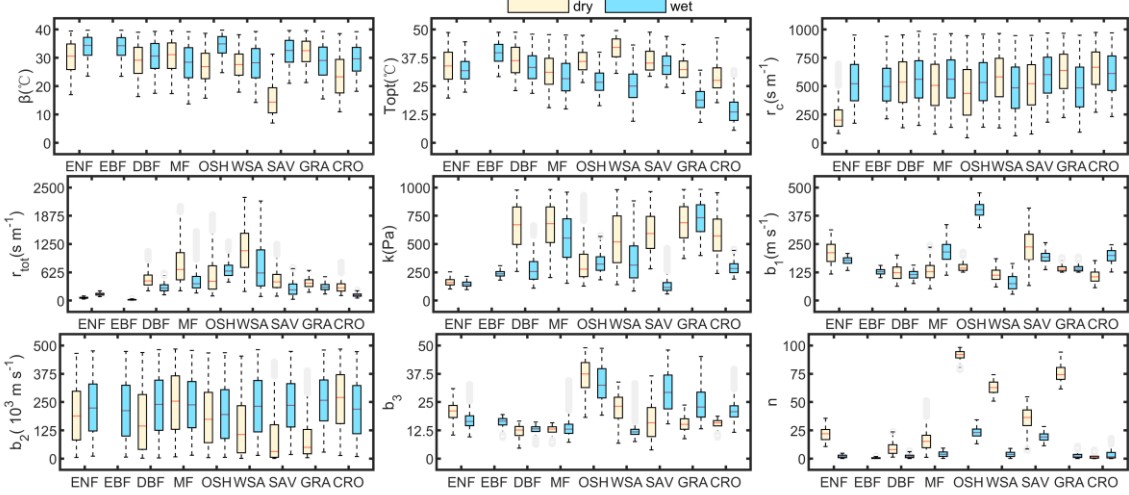

**Figure 4: Boxplots of the parameter posterior distributions with 95% high probability across diverse land cover types and climate zones. The red line represents the median and the gray dots represent outliers.**





In general, all parameters exhibited considerable variability across scenarios, reflecting the diverse responses of different vegetation and soil to the ET process in diverse climates. Among all biome-and-climate-specific parameters, the parameter $n$, which quantifies the soil moisture constraint on vegetation transpiration, warrants further discussion. In all land cover types except EBF and CRO, $n$ is noticeably higher in dry zones compared to wet zones, indicating stronger soil moisture constraints

in dry regions. In almost all wet zones, except for OSH and SAV, $n$ is close to 0, suggesting negligible soil moisture constraints. Even wet OSH and wet SAV represent weak constraint levels, with $n$ of 23.1 and 19.8, respectively. The $n$ values vary widely in dry zones, with the highest 92.9 of OSH, suggesting that only 7.1% of the samples are not stressed by soil moisture. The lowest $n$ in dry zones is 0 of CRO, suggesting that the cropland ecosystem is still not affected by soil moisture even in dry zones. This is because cropland ecosystem is generally subject to regular irrigation schedules, which ensure sufficient water

supply for crop growth and ET dissipation, while the identification of climate zones only considers precipitation and potential ET without accounting for irrigation. Further discussion on uncertainties related to cropland moisture constraint is available in Sect. 4.5. Compared to shrublands and grasslands, the soil moisture constraint of forest towers is generally low. This may partially be attributed to the deep root system of forests, which extends beyond the monitoring coverage of the surface layer. Even so, the surface soil moisture still captures a certain level of moisture constraints on forest ET, with $n$ values of 24.9, 7.8,

and 16.0 for dry ENF, dry DBF, and dry MF, respectively.

In each specific scenario, the parameter $n$ is treated as a constant. Therefore, by combining global patterns of land cover types and climate zones, the spatial patterns of $n$ are plotted in Fig. 5, revealing distinct gradients dominated by surface land cover and climate. The regions where $n$ value exceeds 80, 50, and 20 account for 10.2%, 36.0%, and 44.8% of the global vegetation area, respectively. The regions with strong constraints are primarily located in the Patagonian Plateau of South America, the

western United States, central and southern Africa, central Asia, Oceania, and parts of the pan-Arctic where dry shrublands and grasslands are typically dominant. In contrast, regions with weak or no moisture constraints are concentrated in forests, croplands, and most other wet areas.

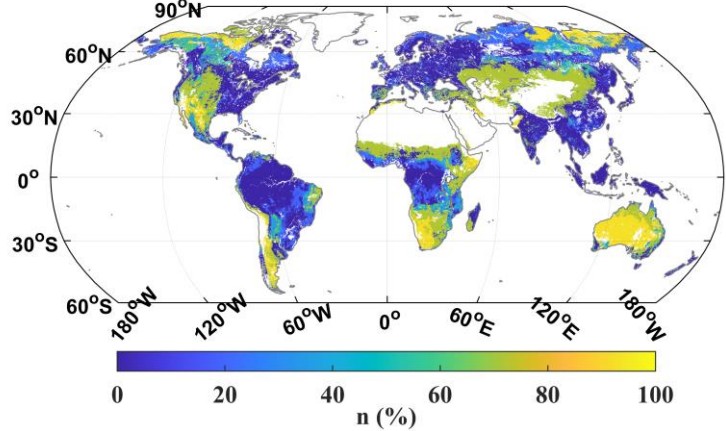

**Figure 5: Global map of the parameter $n$ on vegetation areas covering 9 land cover types.**



The $g_0$~NDVI curves, determined by parameters $b_1$, $b_2$, and $b_3$, are plotted in Fig. 6. Despite sharing a similar sigmoid-type structure, these curves exhibit distinct biological and climatic differences, reflecting variations in leaf traits and physiology. In ENF, DBF, SAV, and GRA, the curves for dry and wet zones are largely consistent, indicating that vegetation stomata respond similarly to NDVI, regardless of climate. However, in some specific NDVI intervals, the corresponding $g_0$ in dry zones is notably higher than that in wet zones for MF, OSH, WSA, and CRO types. One explanation is that this is caused by the actual

response mechanism of vegetation stomata under different climates, indicating that even in the same ecosystem, different climates have an essential impact on the maximum stomatal conductance. More importantly, some vegetation in dry zones is subject to strong moisture constraints from VPD and soil moisture simultaneously. In these cases, the linear structure of constraints in our scheme may reinforce the moisture constraint, which is then compensated for in the potential stomatal conductance response curve.

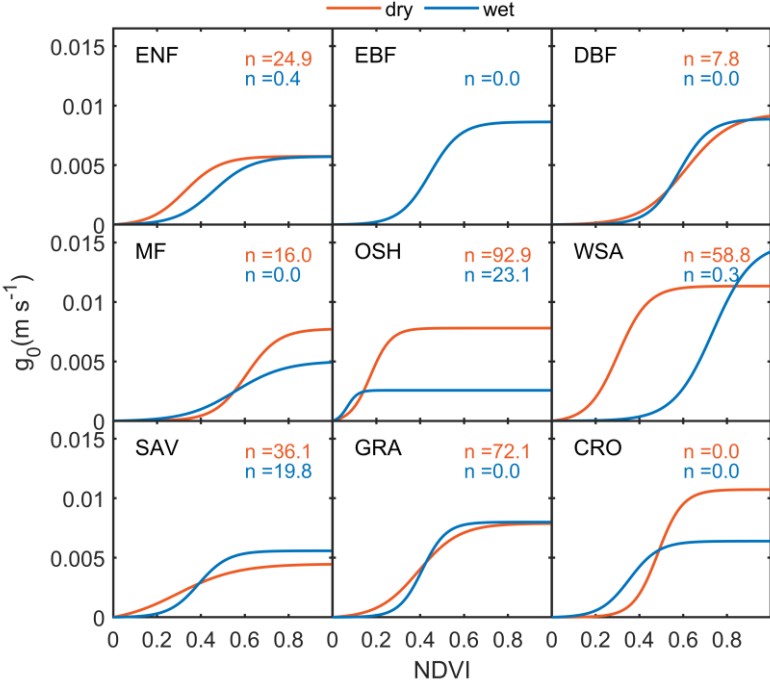


**Figure 6: The sigmoid-type response curve between $g_0$ and NDVI. The colors represent climate zones and *n* is the parameter to control soil moisture constraints.**

### 4.2 Performance related to flux towers

We estimated the daily ET for 106 flux towers using the optimized parameters of the P-LSHv2 algorithm and then compared

the results with ET measurements from flux towers. Due to the limited samples of flux towers in each scenario, the samples were not divided into calibration and validation groups. Instead, all samples were used for parameter calibration (i.e. calibration mode) to evaluate the algorithm performance. The ET estimates of P-LSHv2 and P-LSHv1 were evaluated at each flux tower,

as illustrated in Fig. 7. Overall, the P-LSHv2 algorithm is significantly upgraded. The average RMSE across 106 flux towers drops from 0.79 mm d$^{-1}$ to 0.67 mm d$^{-1}$, a decrease of 15.2%. Similarly, the average NSE increases from 0.43 to 0.58, while

the average $R^2$ sees an increase from 0.62 to 0.67. Among various land cover types, DBF, MF, OSH, WSA, and SAV experienced the most pronounced improvements, with RMSE reduction exceeding 0.15 mm d$^{-1}$, NSE rise level exceeding 0.15, and $R^2$ rise level ranging from 0.03 to 0.17.

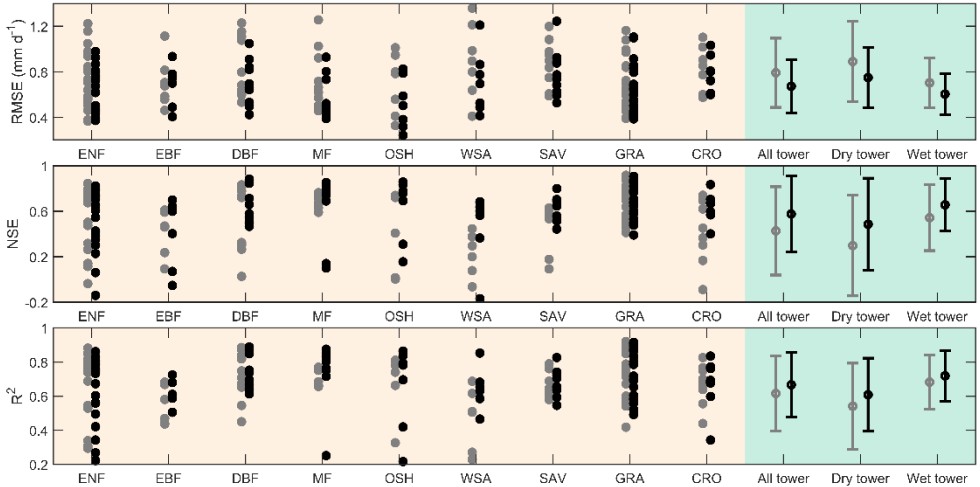

**Figure 7: Comparison of ET estimations from the P-LSHv2 algorithm (black circles) and the P-LSHv1 algorithm (gray circles) at**
**106 flux towers. Each circle represents a flux tower and the classified statistical values are plotted in the right panels. RMSE: root mean square error; NSE: Nash-Sutcliffe efficiency coefficient; $R^2$: coefficient of determination.**

The differences between the two algorithms are more distinct when comparing ET estimations in wet and dry zones. On one hand, the ET estimation in dry zones is generally worse than that in wet zones, the average RMSE of the P-LSHv1 algorithm is 0.89 mm d$^{-1}$ and 0.70 mm d$^{-1}$ (dry zone versus wet zone), and the average RMSE of the P-LSHv2 algorithm is 0.75 mm d$^{-1}$

and 0.60 mm d$^{-1}$ (dry zone versus wet zone). This demonstrates that the physical ET process in dry zones is generally more challenging to simulate than in wet zones. On the other hand, the rising level of the P-LSHv2 estimation in dry zones is higher than that in wet zones. Among all 106 towers, the dry towers show an RMSE decrease of 0.14 mm d$^{-1}$, an NSE increase of 0.19, and an $R^2$ increase of 0.07, compared to RMSE reductions of 0.10 mm d$^{-1}$, NSE improvements of 0.12, and $R^2$ gains of 0.04 in wet zones. We selected several typical dry towers to compare daily ET estimations between two algorithms, as shown

in Fig. 8. At these towers, particularly US-Me6, US-Whs, and US-SRM, the ET estimations significantly improved during water-limited periods, with low values effectively constrained. The three towers represent ENF, OSH, and WSA, respectively, with high $n$ values (24.9, 92.9, and 58.8) and heavy moisture constraints, which further demonstrates the reliability of moisture constraint scheme in P-LSHv2 algorithm. These findings suggest that although the new algorithm has been upgraded in both zones, the soil moisture constraint plays a more essential role in ET simulation in dry zones than in wet zones.

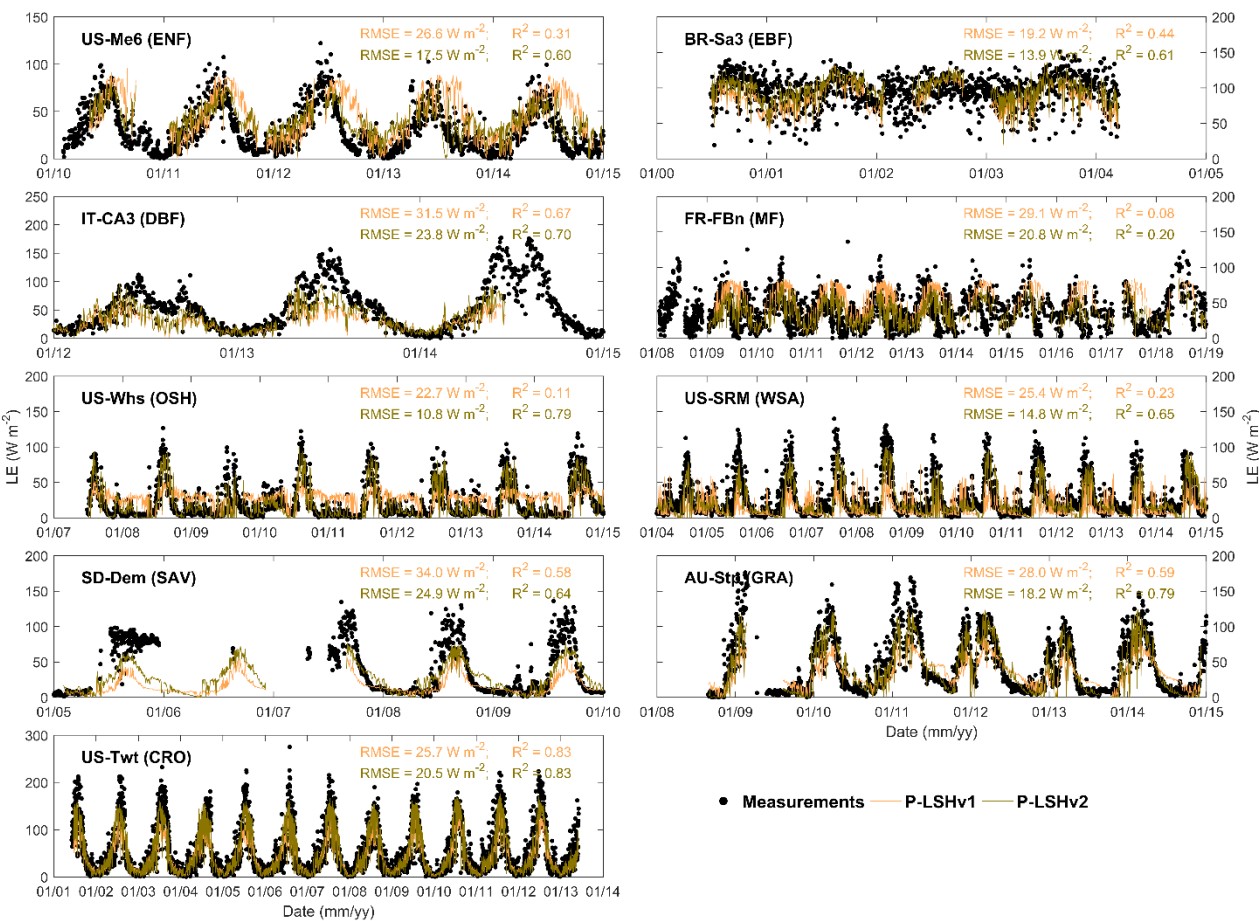


**Figure 8: Time series of daily measured and modelled latent heat flux (LE: W m$^{-2}$) using the P-LSHv2 and P-LSHv1 algorithms for typical dry towers.**

Due to limited samples, all tower measurements for each scenario, covering the entire period of available records, were used for parameter calibration in the preceding parameter characteristic analysis and algorithm evaluation. To further illustrate the

robustness of the P-LSHv2 algorithm and its parameter optimization scheme, we designed another controlled trial using the leave-one-out cross-validation method. In this trial, daily ET measurements from all flux towers within each scenario were randomly split into two groups—one for model calibration and the other serving as the reference truth value for validation using the optimized parameters. This process was repeated twice to ensure that both groups were used both for calibration and validation. The pink and dark symbols in Fig. 9 almost overlap, representing calibration and cross-validation modes

respectively, indicating that the performance of the P-LSHv2 algorithm in cross-validation mode is nearly identical to that in the calibration mode. When rounded to two decimal places, no differences are observed compared to the calibration mode. The largest difference is observed in wet WSA types, where the RMSE difference is only 0.014 mm d$^{-1}$. These results confirm





that the P-LSHv2 algorithm outperforms the P-LSHv1 algorithm in both the calibration and cross-validation modes, demonstrating its stability and robustness in the calibration mode across all climatic and land cover types.

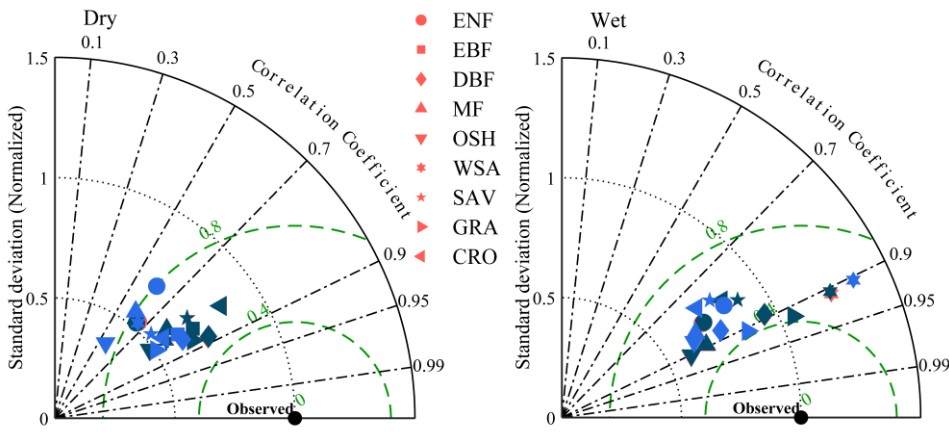


**Figure 9: Taylor diagram evaluating the P-LSHv2 and P-LSHv1 algorithms across all scenarios. Different symbols represent different land cover types. The blue, pink, and dark brown represent the ET estimation statistics from the P-LSHv1 algorithm, the P-LSHv2 algorithm in calibration mode, and the P-LSHv2 algorithm in cross-validation mode, respectively.**

**4.3 Multiscale comparison of the P-LSHv2 results with the other remote sensing products at the basin scale**

The P-LSHv2 algorithm is further evaluated with other global remote sensing ET datasets. Since these datasets are raster-based and derived from remote sensing and reanalysis, their evaluation against flux towers typically encounters issues related to spatial scale inconsistency and upscaling accuracy. Therefore, the reconstructed values from the water balance method ($ET_{recon}$) at the basin scale are selected as the benchmark to assess the performance of various datasets at monthly, annual, and multiyear levels.

**4.3.1 Comparison of ET remote sensing products across the CONUS**

We evaluated gridded ET estimation of 592 basins across CONUS from the P-LSHv2 algorithm, using the reconstructed values from the water balance method as the benchmark. The results were also compared with the P-LSHv1 algorithm and two mainstream products: GLEAM and PML.

At the monthly level, the ET estimation from P-LSHv2 agrees well with $ET_{recon}$, accounting for 71% of the variation, with an
RMSE of 21.4 mm month$^{-1}$ and an NSE of 0.71 (Fig. 10). Compared to P-LSHv1, the RMSE value drops by 7.3%. Moreover, the performance of P-LSHv2 is comparable to that of GLEAM and PML, with acceptable agreement considering the large number of basins and the relatively fine temporal resolution.

Furthermore, we aggregated various monthly ET estimations on an annual basis. On the annual level, the ET estimation from P-LSHv2 upgrades significantly from its predecessor, with an RMSE dropping from 114.4 mm yr$^{-1}$ to 104.4 mm yr$^{-1}$, a decrease





of 8.7% (Fig. 10). Particularly in arid areas (i.e., the basin average AI < 0.65), the P-LSHv2 results effectively correct the overestimation seen in P-LSHv1, reducing the deviation from -44.1 mm yr$^{-1}$ to 27.2 mm yr$^{-1}$. In the driest basins (AI < 0.3), the deviation is reduced even further, from -64.2 mm yr$^{-1}$ to 13.8 mm yr$^{-1}$. These improvements highlight the efficacy of the soil moisture constraints in P-LSHv2, especially in modelling water availability in arid regions. From the perspective of RMSE, NSE, and R$^2$, the ET estimation from P-LSHv2 is also better than that from GLEAM and PML (Fig. 10), with an RMSE of

104.4 mm yr$^{-1}$, an NSE of 0.77 and an R$^2$ of 0.80.

As for the multiyear average scale, the P-LSHv2 results have an RMSE value of 80.4 mm yr$^{-1}$, a decrease of 13.2% compared to its predecessor. In regions with high ET density (around 400 mm yr$^{-1}$), both P-LSHv2 and GLEAM, which account for soil moisture constraints, deliver accurate ET estimates, while PML and P-LSHv1 tend to overestimate, which is an understandable result of their failure to account for soil moisture constraints. In mid-to-high ET intervals (500 mm yr$^{-1}$ to 1000 mm yr$^{-1}$),

GLEAM shows higher uncertainty than P-LSHv2, as reflected in its lower NSE and R$^2$ values, indicating the significant impact of different soil moisture constraint schemes on ET estimation.



**Figure 10: Comparison of monthly, yearly, and multiyear average ET derived from four remote sensing datasets (ET$_{RS}$) and ET$_{recon}$ on 592 basins of CONUS. Colors are used to represent density, with red representing high density and blue representing low density.**

The P-LSHv2 ET estimation also captures spatial gradients across CONUS, with a general relative deviation of –5.7%, significantly lower than the 8.8% deviation of P-LSHv1. About 95.5%, 78.0%, and 35.7% of the area (weighted by size) have

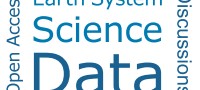



relative deviations within ±30%, ±15%, and ±5%, respectively. Only about 5% of the area experiences underestimation or overestimation beyond 30%. Considerable underestimations are scattered across the Colorado basin, California basin, and parts of the northeast and northwest, while overestimations are concentrated in the West Gulf and North Central basins (Fig.

11). These deviations are likely influenced by reservoir control and interbasin water transfers, which are not accounted for in the model (Wan et al., 2015). Additionally, uncertainties in input data such as precipitation (Daly et al., 2008) and water storage (Landerer and Swenson, 2012) contribute to discrepancies in ET reconstruction. Despite these localized deviations, the general close spatial patterns and minimal differences indicate the high reliability and robustness of P-LSHv2 across CONUS.

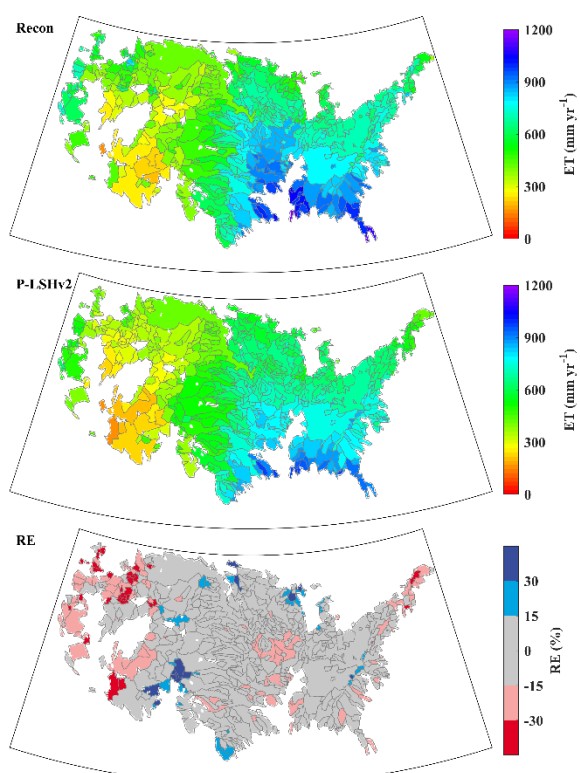

**Figure 11: Spatial patterns of multiyear average ET from ET$_{recon}$ and P-LSHv2 and their difference (RE) on 592 basins of CONUS.**

**4.3.2 Comparison of ET remote sensing products across global large basins**

Apart from CONUS, we also investigated the differences between various remote sensing ET datasets across 32 large basins around the world. Given that the ET$_{recon}$ of these basins spans 1984 to 2006, the shorter PML dataset, which starts from 2002, is excluded.

On the monthly level, GLEAM, P-LSHv1, and P-LSHv2 all accurately capture seasonal ET variations across these basins (Fig. 12), with RMSE values below 15 mm month$^{-1}$ and R$^2$ values exceeding 0.89. Among them, P-LSHv2 achieves the lowest RMSE (9.8 mm month$^{-1}$), improving accuracy by 22.8% over P-LSHv1 and aligning more closely with ET$_{recon}$ than GLEAM.

Earth System
Science
Data

At the annual scale, P-LSHv2 demonstrates a 34.6% improvement in accuracy over P-LSHv1 and shows better agreement with $ET_{recon}$ than GLEAM. Both GLEAM and P-LSHv1 tend to overestimate ET in these 32 basins (Fig. 12), with respective relative

deviations of -23.4% and -20.0%, whereas P-LSHv2 reduces the deviation to -10.9%, indicating that soil moisture constraint incorporated in our algorithm plays an essential role in correcting the overestimation. Compared to the other two datasets, P-LSHv2 particularly agrees well in the low-value intervals below 300 mm yr$^{-1}$ and the high-value intervals above 800 mm yr$^{-1}$. The patterns of multiyear (from 1984 to 2006) average ET estimations of each dataset are similar to those of the annual basis, and P-LSHv2 has lower RMSE, higher NSE, and higher R$^2$ than the other datasets (Fig. 12). These comparisons confirm that

P-LSHv2 performs well in capturing ET patterns across various climate zones and ecosystems around the world.

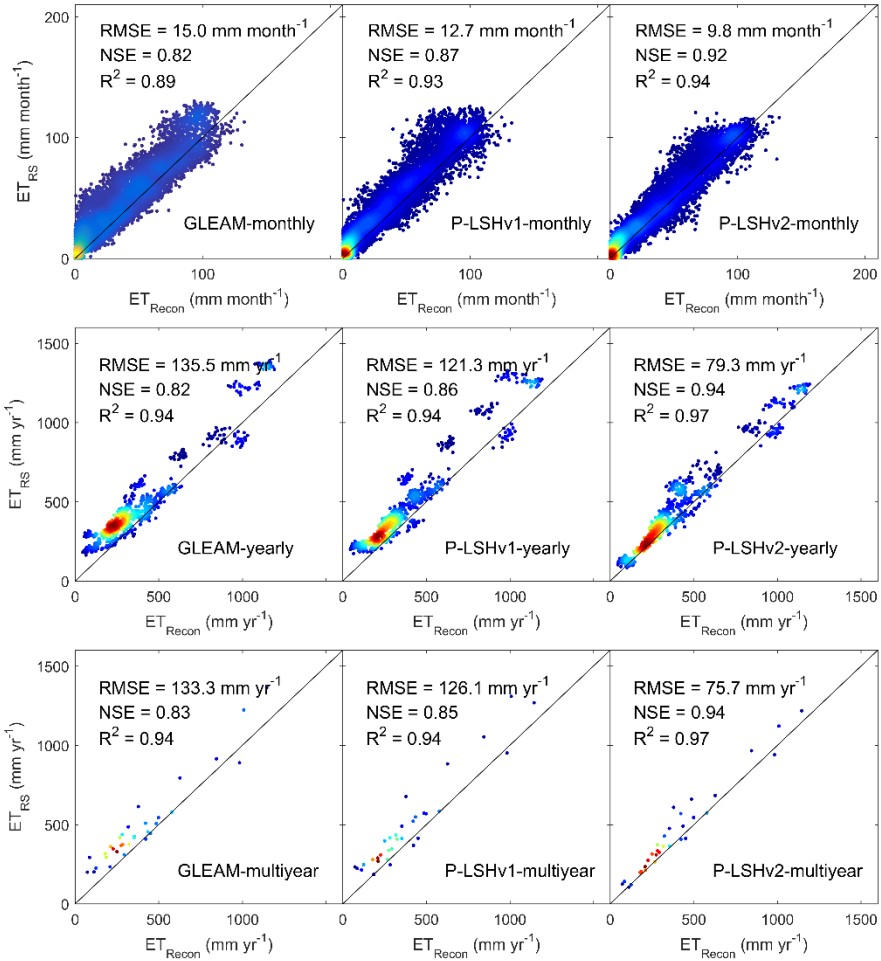

**Figure 12: Comparison of monthly, yearly, and multiyear average ET derived from three remote sensing datasets (ET$_{RS}$) and ET$_{recon}$ on 32 global large basins.**

The spatial patterns of deviations from different datasets vary significantly across 32 basins (Fig. 13). GLEAM and P-LSHv1

show relative deviations exceeding 20% in 19 and 20 basins, respectively, while P-LSHv2 has deviations above 20% in only



10 basins. In basins such as the Amazon, Parana, Mississippi, Niger, Danube, Indus, and Murray-Darling, the deviations of the three datasets all fall within ±20%, demonstrating their strong capability in representing ET budgets at large basin scales. However, some overestimation persists across datasets, notably in the Yukon, Nile, Ural, Indigirka, Amur, and Yangtze basins. This overestimation may be attributed to uncertainties in the water balance model. Since the benchmark data extend to the pre-

GRACE era, Pan et al. (2012) drove the Variable Infiltration Capacity hydrological model to simulate water storage changes, introducing potential errors despite employing Kalman filter techniques to correct for balance errors. Despite some uncertainties, P-LSHv2 has better agreement with $ET_{recon}$ in most basins and shows better accuracy than other remote sensing datasets.

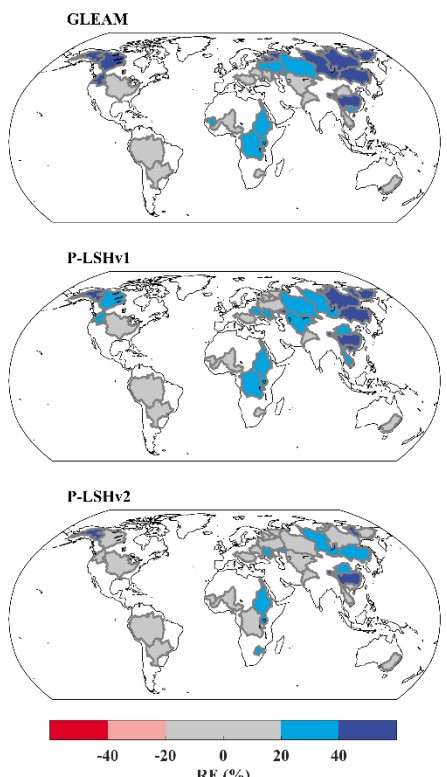

**Figure 13: Spatial patterns of relative deviations of multiyear average ET from three remote sensing datasets on 32 global large basins.**

## 4.4 Global ET patterns

We calculated the global daily terrestrial ET at a spatial resolution of 1/12° from 1982 to 2023 using the improved P-LSHv2 algorithm, excluding permanent snow and ice. The global multiyear average ET reveals distinct regional variations and

latitudinal gradients (Fig. 14). The area-weighted global terrestrial multiyear average ET of P-LSHv2 is 524.8 mm yr$^{-1}$, lower than the 562.2 mm yr$^{-1}$ estimated by P-LSHv1. In dry and wet zones, the area-weighted multiyear averages of P-LSHv2 are



358.8 mm yr$^{-1}$ and 761.4 mm yr$^{-1}$, respectively, both lower than the corresponding estimates of 383.8 mm yr$^{-1}$ and 816.4 mm yr$^{-1}$ from P-LSHv1. The spatial patterns between the two datasets are generally similar, with deviations between P-LSHv2 and P-LSHv1 falling within ± 100 mm yr$^{-1}$ for 56.6% of area-weighted regions. In contrast, 29.5% of the area shows a negative
deviation below -100 mm yr$^{-1}$ and 14.0% of the area shows a positive deviation above 100 mm yr$^{-1}$. The overestimations of P-LSHv1 are particularly concentrated in regions such as the Andes Mountains and Patagonia Plateau of South America, southern North America, and central and southern Africa. These areas are typically composed of OSH, WSA, and SAV biomes, where P-LSHv2 accounts for substantial soil moisture constraints. The higher estimates of P-LSHv2 are located in open water bodies (Great Lakes and the Caspian Sea), bare soils (Sahara Desert, Arabian Peninsula, and central Asia), and some grasslands in
wet-dry boundary regions of the Southern hemisphere.

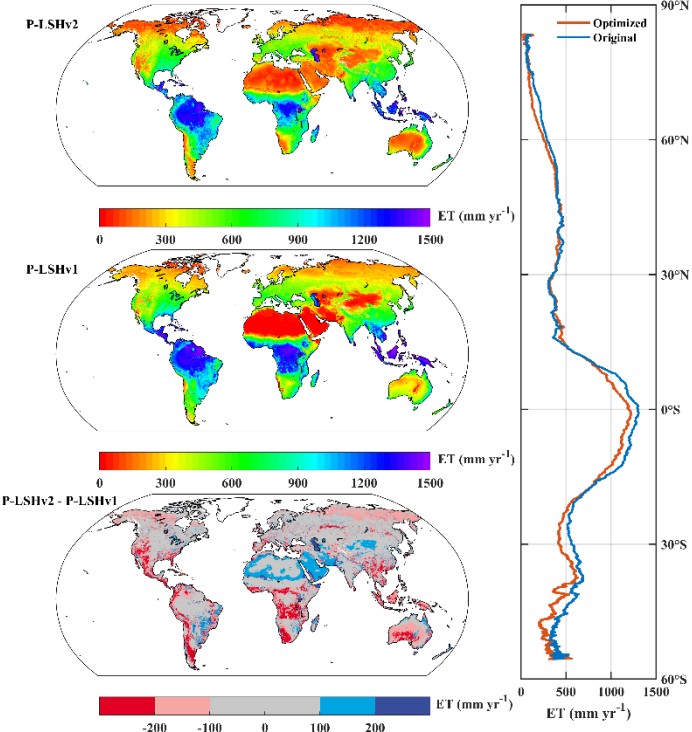

**Figure 14: Global map of multiyear average ET from P-LSHv2 (1982-2023) and P-LSHv1 (1982-2015), as well as their differences and latitudinal profiles.**

In terms of temporal trends, the P-LSHv2 dataset follows a similar trajectory to its predecessor, both exhibiting significant
upward trends (Fig. 15). However, the linear upward trend of P-LSHv2 from 1982 to 2015 is 0.69 mm yr$^{-2}$ ($p$<0.001), slightly lower than that of P-LSHv1, with a trend of 0.87 mm yr$^{-2}$ ($p$<0.001). The trend of P-LSHv2 is comparable to PML (0.68 mm yr$^{-2}$), and higher than GLEAM (0.38 mm yr$^{-2}$). In addition, the decline in net radiation since 2016 has suppressed the upward ET trend, resulting in an increase of only 0.46 ($p$<0.001) mm yr$^{-2}$ from 1982 to 2023.

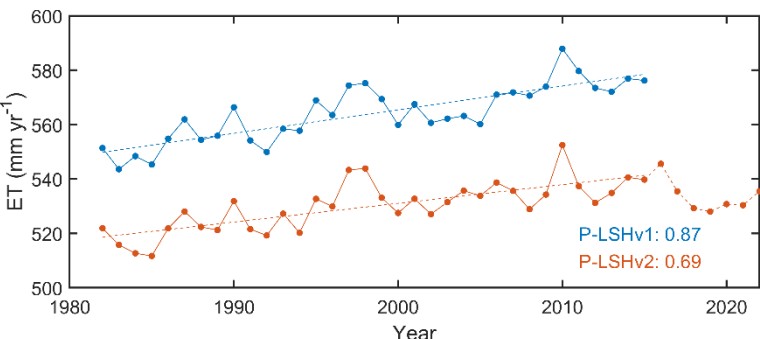

**Figure 15: Annual global ET estimates derived from P-LSHv1 and P-LSHv2.**

## 5 Discussion

Although the P-LSHv2 dataset has good accuracy at multiple temporal and spatial levels around the world, there are still some limitations and uncertainties of the algorithm that are worth discussing.

First, the interaction between soil moisture and ET has not been explicitly considered in this study. We have used existing soil moisture data as an input to estimate ET, without simulating the feedback loop between them. This is primarily due to the focus of this study on assessing the impact of soil moisture on ET estimations. However, it is recognized that SM and ET interact dynamically, and this is a limitation of the current approach. Future work could incorporate the interaction between soil moisture and ET to improve the model's representation of soil water dynamics.

In addition, the soil moisture data used in this study are derived from the GLDAS Noah land surface model rather than satellite remote sensing products. This choice is due to the challenges in obtaining over 40 years of continuous global remote sensing data. The GLDAS Noah model effectively captures seasonal variations and anomalies in soil moisture (Chen and Yuan, 2020; Spennemann et al., 2020) and performs well in ET moisture constraints (Feng et al., 2023). We used data from versions 2.0 and 2.1, covering the period from 1982 to 2023, without applying data fusion, as they originate from the same model and data source. In the P-LSHv2 algorithm, we used surface soil moisture as a moisture constraint surrogate for the entire soil layer. On the one hand, the current satellite remote sensing retrievals struggle to capture soil moisture characteristics of deep layers, and uncertainties associated with deep soil moisture from reanalysis or land surface models are generally greater than those from surface layers. On the other hand, the root zone depth of most tree species tends to be relatively shallow (Flo et al., 2022; Tumber-Dávila et al., 2022), and correlations between soil moisture thresholds at varying depths have been reported (Fu et al., 2022b). Regardless, further model refinement and testing of the P-LSHv2 algorithm could be conducted when high-quality root zone soil moisture datasets become available.

Apart from the forcing data used for ET estimation, there are still certain uncertainties in other ET observations or reconstructions used for algorithm calibration and verification. For instance, the heat flux measurements from eddy covariance towers may suffer from energy imbalance due to the complex circumstances of wind, footprint, and sampling variability



(Wilson et al., 2002). However, it has been shown that there is little difference between daily latent heat flux without the energy
525    closure correction and after the correction (He et al., 2022), and the uncorrected latent heat flux is also widely used in the
evaluation of ET models (Mu et al., 2011; Zhang et al., 2010). As for the benchmark data from 592 basins across CONUS and
32 global large basins, both introduced water storage change data from land surface models to either downscale or extend
temporal coverage, which leads to some uncertainty in small basins or extended years. In the future, further enhancing the
reliability of flux tower observations and water balance reconstructions will promote the development of ET algorithms and
models.

Interestingly, our analysis revealed that dry zones generally experience higher soil moisture constraints than wet zones across
most land cover types, with cropland being a notable exception. Contrary to other scenarios where transpiration in dry zones
is severely limited by soil moisture, flux tower data indicated that the parameter $n$ of cropland ecosystem was 0 for both dry
and wet zones, suggesting that neither was water-limited. Further investigation into five dry cropland towers showed that three
of them (US-Ne2, US-Twt, and IT-CA2) were subject to irrigation exceeding 50 mm yr$^{-1}$, a factor not accounted for in our
climate zone classification. According to the global irrigation dataset from Zhang et al. (2022a), climate zones, and land cover
type data used in this study, approximately 56.5% of global dry cropland is sufficiently irrigated, primarily located in eastern
China, India, and central North America. Understanding how to accurately quantify the unique characteristics of irrigated areas
within the soil moisture constraint framework presents a compelling avenue for future ET simulation research.

**6 Data availability**

All input data used in this study are freely available (see section 3). The P-LSHv2 ET dataset is freely available at the National
Tibetan Plateau Data Center (https://doi.org/10.11888/Terre.tpdc.301969, (Feng Jin, 2025)). Daily, monthly, and yearly
datasets are provided. The dataset is published under the Creative Commons Attribution 4.0 International (CC BY 4.0) license.

**7 Conclusion**

In this study, we incorporated a new global soil moisture constraint scheme into ET estimation and developed a new version
of remote sensing ET algorithm (P-LSHv2). The biome- and climate-specific parameters and soil moisture constraint levels
were calibrated using 106 flux towers around the world. The calibration and cross-validation modes at the flux towers showed
that the new algorithm is robust, and the improvement in the towers' ET simulation for dry zones is higher than that for wet
zones. The investigation of flux towers also quantified the moisture constraint levels across diverse land cover types and
climate zones, revealing that constraints in dry zones generally surpass those in wet zones, particularly for dry OSH, GRA,
and SAV biomes. To evaluate the P-LSHv2 algorithm, we employed two ET benchmarks reconstructed using water balance
methods, assessing performance across multiple temporal scales at the basin level. The P-LSHv2 estimation effectively
captures the impact of soil moisture anomalies on ET, outperforming P-LSHv1, GLEAM, and PML datasets. Consequently,

the P-LSHv2 algorithm provides a more realistic representation of ET processes, and the overall quality of the dataset has been improved, especially in areas where available water is limited. Based on the P-LSHv2 algorithm, a reliable and continuous long-term global ET dataset spanning from 1982 to 2023 has been generated, which contributes to the global assessment of ET climatology and a better understanding of terrestrial water and energy cycle dynamics under climate change conditions.

**Author contribution**

JF and KZ conceived the idea and designed the research. JF and HZ performed the calculation. JF, KZ, LC, YL conducted the analysis. All authors contributed to the results, discussion, and manuscript writing.

**Acknowledgments**

This study was supported by National Key Research and Development Program of China (2023YFC3006505), Fundamental Research Funds for the Central Universities of China (B240203007), and the fund of National Key Laboratory of Water Disaster Prevention (524015222, 2024491611). We gratefully acknowledge Professor Ming Pan from University of California 565 San Diego, for his help in providing the reconstructed ET estimates of global 32 basins. The global P-LSHv2 ET dataset and the code of the P-LSHv2 algorithm are available upon request from the corresponding author. The authors declare no conflict of interest.





## Appendix A

**Table A1.** Details of 106 flux towers used for calibration and validation of ET algorithm.

| Site code | Site name | Latitude | Longitude | IGBP | Years | Network | Arid Index | Climate Zone |
|---|---|---|---|---|---|---|---|---|
| **CA-Man** | Manitoba - Northern Old Black Spruce (former BOREAS Northern Study Area) | 55.880 | -98.481 | ENF | 1994-2008 | FLUXNET | 0.72 | wet |
| **CA-NS2** | UCI-1930 burn site | 55.906 | -98.525 | ENF | 2001-2005 | FLUXNET | 0.72 | wet |
| **CA-NS3** | UCI-1964 burn site | 55.912 | -98.382 | ENF | 2001-2005 | FLUXNET | 0.72 | wet |
| **CN-Qia** | Qianyanzhou | 26.741 | 115.058 | ENF | 2003-2005 | FLUXNET | 1.21 | wet |
| **DE-Lkb** | Lackenberg | 49.100 | 13.305 | ENF | 2009-2013 | FLUXNET | 1.69 | wet |
| **DE-Obe** | Oberbärenburg | 50.787 | 13.721 | ENF | 2008-2014 | FLUXNET | 1.03 | wet |
| **DE-Tha** | Tharandt | 50.962 | 13.565 | ENF | 1996-2014 | FLUXNET | 0.86 | wet |
| **FI-Hyy** | Hyytiala | 61.847 | 24.295 | ENF | 1996-2014 | FLUXNET | 1.04 | wet |
| **FI-Sod** | Sodankylä | 67.362 | 26.638 | ENF | 2001-2014 | FLUXNET | 0.96 | wet |
| **FR-LBr** | Le Bray | 44.717 | -0.769 | ENF | 1996-2008 | FLUXNET | 0.87 | wet |
| **NL-Loo** | Loobos | 52.167 | 5.744 | ENF | 1996-2014 | FLUXNET | 1.09 | wet |
| **RU-Fyo** | Fyodorovskoye | 56.462 | 32.922 | ENF | 1998-2014 | FLUXNET | 0.95 | wet |
| **US-Blo** | Blodgett Forest | 38.895 | -120.633 | ENF | 1997-2007 | FLUXNET | 0.84 | wet |
| **US-Me2** | Metolius mature ponderosa pine | 44.452 | -121.557 | ENF | 2002-2014 | FLUXNET | 0.65 | wet |
| **AU-ASM** | Alice Springs | -22.283 | 133.249 | ENF | 2010-2014 | FLUXNET | 0.12 | dry |
| **CA-SF1** | Saskatchewan - Western Boreal, forest burned in 1977 | 54.485 | -105.818 | ENF | 2003-2006 | FLUXNET | 0.61 | dry |
| **CA-SF2** | Saskatchewan - Western Boreal, forest burned in 1989 | 54.254 | -105.878 | ENF | 2001-2005 | FLUXNET | 0.59 | dry |
| **IT-Lav** | Lavarone | 45.956 | 11.281 | ENF | 2003-2014 | FLUXNET | 0.64 | dry |
| **US-GBT** | GLEES Brooklyn Tower | 41.366 | -106.240 | ENF | 1999-2006 | FLUXNET | 0.44 | dry |
| **US-GLE** | GLEES | 41.367 | -106.240 | ENF | 2004-2014 | FLUXNET | 0.44 | dry |
| **US-Me6** | Metolius Young Pine Burn | 44.323 | -121.608 | ENF | 2010-2014 | FLUXNET | 0.50 | dry |
| **US-NR1** | Niwot Ridge Forest (LTER NWT1) | 40.033 | -105.546 | ENF | 1998-2014 | FLUXNET | 0.43 | dry |
| **AU-Cum** | Cumberland Plains | -33.613 | 150.723 | EBF | 2012-2014 | FLUXNET | -* | - |
| **AU-Tum** | Tumbarumba | -35.657 | 148.152 | EBF | 2001-2014 | FLUXNET | - | - |
| **AU-Whr** | Whroo | -36.673 | 145.029 | EBF | 2011-2014 | FLUXNET | - | - |
| **BR-Sa3** | Santarem-Km83-Logged Forest | -3.018 | -54.971 | EBF | 2000-2004 | FLUXNET | - | - |
| **CN-Din** | Dinghushan | 23.173 | 112.536 | EBF | 2003-2005 | FLUXNET | - | - |
| **GF-Guy** | Guyaflux (French Guiana) | 5.279 | -52.925 | EBF | 2004-2014 | FLUXNET | - | - |
| **MY-PSO** | Pasoh Forest Reserve (PSO) | 2.973 | 102.306 | EBF | 2003-2009 | FLUXNET | - | - |



| Site code | Site name | Latitude | Longitude | IGBP | Years | Network | Arid Index | Climate Zone |
|---|---|---|---|---|---|---|---|---|
| DE-Hai | Hainich | 51.079 | 10.453 | DBF | 2000-2012 | FLUXNET | 0.93 | wet |
| FR-Fon | Fontainebleau-Barbeau | 48.476 | 2.780 | DBF | 2005-2014 | FLUXNET | 0.68 | wet |
| US-Ha1 | Harvard Forest EMS Tower (HFR1) | 42.538 | -72.172 | DBF | 1991-2012 | FLUXNET | 1.09 | wet |
| US-UMB | Univ. of Mich. Biological Station | 45.560 | -84.714 | DBF | 2000-2014 | FLUXNET | 0.82 | wet |
| US-UMd | UMBS Disturbance | 45.563 | -84.698 | DBF | 2007-2014 | FLUXNET | 0.82 | wet |
| PA-SPn | Sardinilla Plantation | 9.318 | -79.635 | DBF | 2007-2009 | FLUXNET | 1.60 | wet |
| JP-MBF | Moshiri Birch Forest Site | 44.387 | 142.319 | DBF | 2003-2005 | FLUXNET | 1.53 | wet |
| IT-CA1 | Castel d'Asso 1 | 42.380 | 12.027 | DBF | 2011-2014 | FLUXNET | 0.24 | dry |
| IT-CA3 | Castel d'Asso 3 | 42.380 | 12.022 | DBF | 2011-2014 | FLUXNET | 0.24 | dry |
| ZM-Mon | Mongu | -15.438 | 23.253 | DBF | 2000-2009 | FLUXNET | 0.39 | dry |
| CA-Oas | Saskatchewan - Western Boreal, Mature Aspen | 53.629 | -106.198 | DBF | 1996-2010 | FLUXNET | 0.55 | dry |
| CN-Cha | Changbaishan | 42.403 | 128.096 | MF | 2003-2005 | FLUXNET | 0.77 | wet |
| US-Syv | Sylvania Wilderness Area | 46.242 | -89.348 | MF | 2001-2014 | FLUXNET | 0.95 | wet |
| CA-Gro | Ontario - Groundhog River, Boreal Mixedwood Forest | 48.217 | -82.156 | MF | 2003-2014 | FLUXNET | 0.95 | wet |
| BE-Bra | Brasschaat | 51.308 | 4.520 | MF | 1996-2014 | FLUXNET | 0.95 | wet |
| BE-Vie | Vielsalm | 50.305 | 5.998 | MF | 1996-2014 | FLUXNET | 1.40 | wet |
| DE-Meh | Mehrstedt | 51.275 | 10.655 | MF | 2003-2006 | European Flux | 0.86 | wet |
| AR-SLu | San Luis | -33.465 | -66.460 | MF | 2009-2011 | FLUXNET | 0.21 | dry |
| CZ-Lnz | Lanzhot | 48.682 | 16.946 | MF | 2015-2020 | European Flux | 0.63 | dry |
| FR-FBn | Font-Blanche | 43.241 | 5.679 | MF | 2008-2018 | European Flux | 0.46 | dry |
| IT-Non | Nonantola | 44.690 | 11.091 | MF | 2001-2018 | European Flux | 0.60 | dry |
| CA-NS6 | UCI-1989 burn site | 55.917 | -98.964 | OSH | 2001-2005 | FLUXNET | 0.70 | wet |
| CA-NS7 | UCI-1998 burn site | 56.636 | -99.948 | OSH | 2002-2005 | FLUXNET | 0.69 | wet |
| US-Wi6 | Pine barrens #1 (PB1) | 46.625 | -91.298 | OSH | 2002-2003 | FLUXNET | 0.85 | wet |
| US-Wi7 | Red pine clearcut (RPCC) | 46.649 | -91.069 | OSH | 2005-2005 | FLUXNET | 0.85 | wet |
| AU-TTE | Ti Tree East | -22.287 | 133.640 | OSH | 2012-2014 | FLUXNET | 0.09 | dry |
| CA-SF3 | Saskatchewan - Western Boreal, forest burned in 1998 | 54.092 | -106.005 | OSH | 2001-2006 | FLUXNET | 0.58 | dry |
| US-Whs | Walnut Gulch Lucky Hills Shrub | 31.744 | -110.052 | OSH | 2007-2014 | FLUXNET | 0.15 | dry |
| BR-Npw | Northern Pantanal Wetland | -16.498 | -56.412 | WSA | 2013-2017 | AmeriFlux | 0.87 | wet |
| RU-Zot | Zotino | 60.801 | 89.351 | WSA | 2002-2004 | European Flux | 0.85 | wet |
| AU-Ade | Adelaide River | -13.077 | 131.118 | WSA | 2007-2009 | FLUXNET | 0.62 | dry |
| AU-Gin | Gingin | -31.376 | 115.714 | WSA | 2011-2014 | FLUXNET | 0.32 | dry |
| AU-RDF | Red Dirt Melon Farm, Northern Territory | -14.564 | 132.478 | WSA | 2011-2013 | FLUXNET | 0.39 | dry |
| US-SRM | Santa Rita Mesquite | 31.821 | -110.866 | WSA | 2004-2014 | FLUXNET | 0.19 | dry |



| Site code | Site name | Latitude | Longitude | IGBP | Years | Network | Arid Index | Climate Zone |
|---|---|---|---|---|---|---|---|---|
| US-Ton | Tonzi Ranch | 38.432 | -120.966 | WSA | 2001-2014 | FLUXNET | 0.33 | dry |
| CG-Tch | Tchizalamou | -4.289 | 11.656 | SAV | 2006-2009 | FLUXNET | 1.09 | wet |
| US-LL1 | Longleaf Pine - Baker (Mesic site) | 31.279 | -84.533 | SAV | 2009-2020 | AmeriFlux | 0.83 | wet |
| US-LL2 | Longleaf Pine - Dubignion (Intermediate site) | 31.201 | -84.445 | SAV | 2009-2017 | AmeriFlux | 0.83 | wet |
| US-LL3 | Longleaf Pine - Red Dirt (Xeric site) | 31.269 | -84.479 | SAV | 2009-2017 | AmeriFlux | 0.83 | wet |
| AU-Cpr | Calperum | -34.002 | 140.589 | SAV | 2010-2014 | FLUXNET | 0.12 | dry |
| AU-DaS | Daly River Cleared | -14.159 | 131.388 | SAV | 2008-2014 | FLUXNET | 0.48 | dry |
| AU-Dry | Dry River | -15.259 | 132.371 | SAV | 2008-2014 | FLUXNET | 0.34 | dry |
| SD-Dem | Demokeya | 13.283 | 30.478 | SAV | 2005-2009 | FLUXNET | 0.10 | dry |
| SN-Dhr | Dahra | 15.403 | -15.432 | SAV | 2010-2013 | FLUXNET | 0.14 | dry |
| ZA-Kru | Skukuza | -25.020 | 31.497 | SAV | 2000-2013 | FLUXNET | 0.37 | dry |
| AT-Neu | Neustift | 47.117 | 11.318 | GRA | 2002-2012 | FLUXNET | 1.31 | wet |
| CH-Cha | Chamau | 47.210 | 8.410 | GRA | 2005-2014 | FLUXNET | 1.38 | wet |
| CH-Fru | Früebüel | 47.116 | 8.538 | GRA | 2005-2014 | FLUXNET | 1.63 | wet |
| CH-Oe1 | Oensingen grassland | 47.286 | 7.732 | GRA | 2002-2008 | FLUXNET | 1.37 | wet |
| DE-RuR | Rollesbroich | 50.622 | 6.304 | GRA | 2011-2014 | FLUXNET | 1.38 | wet |
| IT-Tor | Torgnon | 45.844 | 7.578 | GRA | 2008-2014 | FLUXNET | 1.24 | wet |
| NL-Hor | Horstermeer | 52.240 | 5.071 | GRA | 2004-2011 | FLUXNET | 1.09 | wet |
| AU-DaP | Daly River Savanna | -14.063 | 131.318 | GRA | 2007-2013 | FLUXNET | 0.50 | dry |
| AU-Rig | Riggs Creek | -36.650 | 145.576 | GRA | 2011-2014 | FLUXNET | 0.37 | dry |
| AU-Stp | Sturt Plains | -17.151 | 133.350 | GRA | 2008-2014 | FLUXNET | 0.23 | dry |
| CN-Cng | Changling | 44.593 | 123.509 | GRA | 2007-2010 | FLUXNET | 0.32 | dry |
| CN-Dan | Dangxiong | 30.498 | 91.066 | GRA | 2004-2005 | FLUXNET | 0.33 | dry |
| CN-Du2 | Duolun_grassland (D01) | 42.047 | 116.284 | GRA | 2006-2008 | FLUXNET | 0.33 | dry |
| CN-Sw2 | Siziwang Grazed (SZWG) | 41.790 | 111.897 | GRA | 2010-2012 | FLUXNET | 0.15 | dry |
| IT-MBo | Monte Bondone | 46.015 | 11.046 | GRA | 2003-2013 | FLUXNET | 0.55 | dry |
| RU-Ha1 | Hakasia steppe | 54.725 | 90.002 | GRA | 2002-2004 | FLUXNET | 0.56 | dry |
| US-AR1 | ARM USDA UNL OSU Woodward Switchgrass 1 | 36.427 | -99.420 | GRA | 2009-2012 | FLUXNET | 0.30 | dry |
| US-AR2 | ARM USDA UNL OSU Woodward Switchgrass 2 | 36.636 | -99.598 | GRA | 2009-2012 | FLUXNET | 0.30 | dry |
| US-ARb | ARM Southern Great Plains burn site-Lamont | 35.550 | -98.040 | GRA | 2005-2006 | FLUXNET | 0.44 | dry |
| US-ARc | ARM Southern Great Plains control site-Lamont | 35.547 | -98.040 | GRA | 2005-2006 | FLUXNET | 0.44 | dry |
| US-SRG | Santa Rita Grassland | 31.789 | -110.828 | GRA | 2008-2014 | FLUXNET | 0.22 | dry |
| US-Wkg | Walnut Gulch Kendall Grasslands | 31.737 | -109.942 | GRA | 2004-2014 | FLUXNET | 0.17 | dry |



| Site code | Site name | Latitude | Longitude | IGBP | Years | Network | Arid Index | Climate Zone |
|-----------|-----------|----------|-----------|------|-------|---------|-----------|--------------|
| DE-Seh | Selhausen | 50.871 | 6.450 | CRO | 2007-2010 | FLUXNET | 0.81 | wet |
| FR-Gri | Grignon | 48.844 | 1.952 | CRO | 2004-2014 | FLUXNET | 0.69 | wet |
| US-CRT | Curtice Walter-Berger cropland | 41.629 | -83.347 | CRO | 2011-2013 | FLUXNET | 0.74 | wet |
| US-Ro1 | Rosemount- G21 | 44.714 | -93.090 | CRO | 2004-2016 | AmeriFlux | 0.71 | wet |
| US-Mo1 | LTAR CMRB Field 1 (CMRB ASP) | 39.230 | -92.117 | CRO | 2015-2020 | AmeriFlux | 0.75 | wet |
| US-ARM | ARM Southern Great Plains site- Lamont | 36.606 | -97.489 | CRO | 2003-2012 | FLUXNET | 0.49 | dry |
| DE-Geb | Gebesee | 51.100 | 10.914 | CRO | 2001-2014 | FLUXNET | 0.60 | dry |
| US-Ne2 | Mead - irrigated maize-soybean rotation site | 41.165 | -96.470 | CRO | 2001-2013 | FLUXNET | 0.58 | dry |
| IT-CA2 | Castel d'Asso 2 | 42.377 | 12.026 | CRO | 2011-2014 | FLUXNET | 0.24 | dry |
| US-Twt | Twitchell Island | 38.109 | -121.653 | CRO | 2009-2014 | FLUXNET | 0.22 | dry |

*The EBF is no longer categorized into various climate zones, as less than 2% of the global EBF falls within dry zones.





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
