# Peer review of "P-LSHv2: a multi-decadal global daily evapotranspiration dataset enhanced with explicit soil moisture constraints"

_Earth System Science Data, 2025_

## Referee Comment (RC1)

**Comments on Feng et al. 2025 April**

Dr Paul Stuart Blackwell (Geraldton WA)

1. **"a multi-decadal global daily land surface actual evapotranspiration dataset enhanced with explicit soil moisture constraints in remote sensing retrieval"**

Could the title possibly be better expressed, thus-?

"**Global daily evapotranspiration estimated from land surfaces by remote sensing over multiple decades, including explicit soil moisture constraints to remote data retrieval.**"

2. "*We integrated this approach into the process-based land surface 20 ET/heat fluxes algorithm (P-LSH, or P-LSHv1), developing an improved version, P-LSHv2. Using observations from 106 global flux towers, we calibrated biome- and climate-specific parameters and quantified moisture constraints across diverse climates and land cover types. P-LSHv2 achieves notable improvements in ET estimation, with a reduced Root Mean Square Error (RMSE) of 0.67 mm d $^{-1}$ and an increased correlation coefficient (R) of 0.81, outperforming its predecessor, P -LSHv1, particularly in arid regions.*"

A most efficient description of complex processes, but, should (R be $R^2$)?

3. "*Leveraging the P-LSHv2 algorithm, we have produced a long-term global daily ET dataset spanning 1982‑2023, providing a valuable resource for research on terrestrial water and energy cycles and climate change. The dataset is freely available at https://doi.org/10.11888/Terre.tpdc.301969 (Feng Jin, 2025).*"
This is a very generous offer of free access to your data, Jin.
I just question your choice of the word 'Leveraging'. I know I am a 72-year old, old fashioned bloke who still used printed map books to figure out where to drive in the city, but still have a more than adequate mental map of the whole of SW Australia to call on from long years of driving around helping agriculture. But the point is that the word 'Leveraging' primarily reminds me of the very skilled Aussie tyre fitter who I often had to call upon to change a tyre or two on the government car I was driving around. So maybe for the sake of a broad readership of your extensive paper, the word 'employing' may be a more suitable one in these circumstances?  Just a respectful suggestion.

4. "*Due to the water potential gradient between leaf and air, water is transported from soil to vegetation roots, and leaves, and then dissipated into the atmosphere through stomata. Therefore, soil water content serves as the direct water pool for vegetation and regulates the magnitude of water extracted by vegetation roots (Feng et al., 2022; Liu et al., 2020b)*"
This is an eloquent, but oversimplified, physical explanation of evapotranspiration. It requires inclusion of the biological need and purpose of transpiration by plants and the vital role to sensory and growth behaviour that plant root tips play in semi-arid ecologies especially in landscapes with soil types of poor water-holding capacity. This text is extracted from one of the research papers I am developing.

"*Dexter (1986) described the behaviour of plant roots seeking biopores, some concepts have been put forward, such as "trematotropism" and "oxytropism".* **Gregory (2009) summarised that 'Roots grow towards areas of higher water potential ... and that roots could sense a water potential gradient as small as 0.5 MPa mm–1 so that hydroresponsiveness may contribute to both avoidance of drought stress and modifications to root system architecture'.** *This knowledge strongly suggest that the soil profile structure needs some degree of heterogeneity varying from loose structure for ease of root*

*exploration to more dense components (clods or ridges) which allow only slow or little root growth and can retain moisture at higher potential; more readily available at times when growing conditions are drier. In a similar manner, roots seek out some nutrients along gradients of their occurrence in the soil profile as nutrients are supplied to the root surface by mass flow and diffusion'."*

Thus, by logical deduction, the ability of root tips to search out water in the soil profile may have more control on ET that the simple vapour deficit gradient. Additionally, since most of the evaporated water is used to cool leaves on hot afternoons, any undersupply and overheating leads to a breakdown of ET pathways through the plant tissue and a reduction of ET despite a strong VP gradient. Such processes do need to be explained and included in this MS, and there may well be more research of that aspect, since I am not fully up-to date with that research sector.

Comments completed up to page 4.  PSB 2/4/2025

---

## Community Comment (CC1)

**Response to Reviewers' comments**

We sincerely appreciate Dr Paul Stuart Blackwell for the valuable and constructive comments, which will greatly help us improve the quality of our manuscript. We have carefully considered all comments and will revise the manuscript accordingly. The point-to-point responses to the comments and our plans for revision are listed below.

**Replies to Comments:**

1. "a multi-decadal global daily land surface actual evapotranspiration dataset enhanced with explicit soil moisture constraints in remote sensing retrieval"

Could the title possibly be better expressed, thus-?

"Global daily evapotranspiration estimated from land surfaces by remote sensing over multiple decades, including explicit soil moisture constraints to remote data retrieval."

**Response:**

We sincerely thank the reviewer for the thoughtful suggestion regarding the manuscript title. We agree that the proposed version improves fluency and places helpful emphasis on "global daily evapotranspiration." While our original title was indeed longer, it aimed to reflect the dataset's key characteristics and methodological foundation, consistent with the conventions of *Earth System Science Data (ESSD)* data description papers. For reference, similar titles in ESSD include:

- *CAMELE: Collocation-Analyzed Multi-source Ensembled Land Evapotranspiration Data*
- *A global 5 km monthly potential evapotranspiration dataset (1982–2015) estimated by the Shuttleworth–Wallace model*
- A daily and 500 m coupled evapotranspiration and gross primary production product across China during 2000–2020
- A global terrestrial evapotranspiration product based on the three-temperature model with fewer input parameters and no calibration requirement

In consideration of the reviewer's valuable feedback and to better align with ESSD's style and audience expectations, we propose revising the title to:

**P-LSHv2: A multi-decadal global daily evapotranspiration dataset enhanced with explicit soil moisture constraints**

We believe this revised title improves clarity and conciseness while preserving the necessary level of detail and methodological specificity.

2. "We integrated this approach into the process-based land surface 20 ET/heat fluxes algorithm (P-LSH, or P-LSHv1), developing an improved version, P-LSHv2. Using observations from 106 global flux towers, we calibrated biome- and

climate-specific parameters and quantified moisture constraints across diverse climates and land cover types. P-LSHv2 achieves notable improvements in ET estimation, with a reduced Root Mean Square Error (RMSE) of 0.67 mm d-1 and an increased correlation coefficient (R) of 0.81, outperforming its predecessor, P -LSHv1, particularly in arid regions."

A most efficient description of complex processes, but, should (R be  $R^2$ )?

**Response:**

We thank the reviewer for the positive feedback and the insightful question. In this context, we used the **Pearson correlation coefficient (R)** to assess the linear agreement between the estimated and observed evapotranspiration (ET) values across flux towers. Since our focus is on evaluating consistency rather than the proportion of explained variance—as would be the case with the coefficient of determination  $(R^2)$ —we believe that reporting  $\mathbf{R} = \mathbf{0.81}$  is appropriate.

To avoid any potential confusion, we will clarify this explicitly in the revised manuscript. The sentence has been revised as follows:

"...P-LSHv2 achieves notable improvements in ET estimation, with a reduced root mean square error (RMSE) of 0.67 mm d-1 and an increased **Pearson correlation coefficient (R)** of 0.81, indicating strong agreement with flux tower observations. As a result of these improvements, P-LSHv2 outperforms its predecessor, P-LSHv1, particularly in arid regions..."

3. "Leveraging the P-LSHv2 algorithm, we have produced a long-term global daily ET dataset spanning 1982–2023, providing a valuable resource for research on terrestrial water and energy cycles and climate change. The dataset is freely available at https://doi.org/10.11888/Terre.tpdc.301969 (Feng Jin, 2025)."

This is a very generous offer of free access to your data, Jin.

I just question your choice of the word 'Leveraging'. I know I am a 72-year old, old fashioned bloke who still used printed map books to figure out where to drive in the city, but still have a more than adequate mental map of the whole of SW Australia to call on from long years of driving around helping agriculture. But the point is that the word 'Leveraging' primarily reminds me of the very skilled Aussie tyre fitter who I often had to call upon to change a tyre or two on the government car I was driving around. So maybe for the sake of a broad readership of your extensive paper, the word 'employing' may be a more suitable one in these circumstances? Just a respectful suggestion.

**Response:**

We sincerely thank the reviewer for the kind words and for the thoughtful suggestion regarding word choice. We appreciate the perspective on the term "leveraging," and

agree that "employing" may read more naturally and be more widely accessible to a broad readership. To improve clarity and tone, we have revised the sentence as follows:

"**Employing the P-LSHv2 algorithm**, we have produced a long-term global daily ET dataset spanning 1982–2023..."

We are grateful for the reviewer's attention to both language and accessibility, which contributes meaningfully to improving the manuscript.

4. "Due to the water potential gradient between leaf and air, water is transported from soil to vegetation roots, and leaves, and then dissipated into the atmosphere through stomata. Therefore, soil water content serves as the direct water pool for vegetation and regulates the magnitude of water extracted by vegetation roots (Feng et al., 2022; Liu et al., 2020b)"

This is an eloquent, but oversimplified, physical explanation of evapotranspiration. It requires inclusion of the biological need and purpose of transpiration by plants and the vital role to sensory and growth behaviour that plant root tips play in semi-arid ecologies especially in landscapes with soil types of poor water-holding capacity. This text is extracted from one of the research papers I am developing.

"Dexter (1986) described the behaviour of plant roots seeking biopores, some concepts have been put forward, such as "trematotropism" and "oxytropism". Gregory (2009) summarised that 'Roots grow towards areas of higher water potential ... and that roots could sense a water potential gradient as small as 0.5 MPa mm-1 so that hydroresponsiveness may contribute to both avoidance of drought stress and modifications to root system architecture'. This knowledge strongly suggest that the soil profile structure needs some degree of heterogeneity varying from loose structure for ease of root exploration to more dense components (clods or ridges) which allow only slow or little root growth and can retain moisture at higher potential; more readily available at times when growing conditions are drier. In a similar manner, roots seek out some nutrients along gradients of their occurrence in the soil profile as nutrients are supplied to the root surface by mass flow and diffusion'."

Thus, by logical deduction, the ability of root tips to search out water in the soil profile may have more control on ET that the simple vapour deficit gradient. Additionally, since most of the evaporated water is used to cool leaves on hot afternoons, any undersupply and overheating leads to a breakdown of ET pathways through the plant tissue and a reduction of ET despite a strong VP gradient. Such processes do need to be explained and included in this MS, and there may well be more research of that aspect, since I am not fully up-to date with that research sector. **Response:**

We sincerely appreciate the reviewer for the insightful and constructive comments. Your suggestions have significantly enriched our understanding of plant water use strategies and the underlying ecological mechanisms. Below, we summarize the key issues you raised and provide our detailed responses: (1) Transpiration mechanism oversimplified

The explanation of plant transpiration in the manuscript is overly simplified, focusing mainly on physical processes (i.e., water potential gradients) while neglecting physiological drivers and regulatory mechanisms—especially plant responses under drought constraints.

(2) Active root sensing and hydrotropism

Plant roots are not passive in water uptake but actively sense and grow toward water through physiological mechanisms. The ability to detect subtle water potential gradients and directionally grow plays a key role in maintaining transpiration under moisture-constrained conditions.

(3) Soil structure effects on root uptake

Soil structural heterogeneity substantially influences root architecture and water availability. Loose soils promote root exploration, while compacted structures aid in water retention, thereby shaping effective water uptake and transpiration dynamics.

(4) ET limitation under drought conditions

In dryland ecosystems, root water acquisition may exert a more direct control on ET than atmospheric drivers such as vapor pressure deficit (VPD). Under drought, even high VPD may not lead to higher ET due to limited plant access to water.

In response to these points, we will revise the manuscript as follows:

**Response to Point (1):**

We acknowledge that our previous description of transpiration primarily focused on its physical pathway. In the revised manuscript, we will add further explanation of the physiological regulation of transpiration. Specifically:

"In addition to the physical gradient of water potential, plant transpiration is fundamentally driven by biological needs such as nutrient transport, turgor maintenance, and leaf cooling. These physiological functions are tightly regulated and feed back to control stomatal conductance, thereby influencing transpiration dynamics".

**Response to Point (2):**

We will revise the description of root water uptake to clarify that roots are not merely passive structures. Instead, we will emphasize their sensory and active water-seeking behavior. The revised text will include:

"Root tips are capable of sensing subtle gradients in water potential (as low as 0.5 MPa mm-1), exhibiting behaviors such as hydrotropism to actively seek water in heterogeneous soil profiles (Gregory, 2009; Dexter, 1986). Such sensory responses provide a physiological basis for root foraging behavior, which is particularly important for sustaining transpiration under drought conditions.".

**Response to Point (3):**

We agree that soil structural heterogeneity plays a key role in root development and water availability. In our P-LSHv2 algorithm, such heterogeneity is indirectly represented by land cover and climate classifications, which is determined by the parameter n. We opted not to use global soil hydraulic properties due to their high uncertainty, but land cover and climate types provide a feasible proxy for large-scale heterogeneity. We will also mention the potential of incorporating higher-resolution soil hydraulic properties in future work. The following paragraph will be added to the discussion:

"Soil structural heterogeneity plays a crucial role in regulating the distribution and availability of soil water. Looser soil facilitates root penetration, while denser soil can retain water at higher matric potentials, thus extending water availability during dry periods. This spatial variation influences root distribution patterns and overall transpiration rates".

**Response to Point (4):**

We agree that in arid ecosystems, the availability of soil water may limit ET more directly than VPD. This perspective supports our inclusion of explicit soil moisture constraints in the P-LSHv2 algorithm. Our results also indicate that ET in arid ecosystems is highly sensitive to soil water availability. We will add the following explanation in the manuscript:

"In arid and semi-arid regions, even under high atmospheric demand (i.e., high VPD), the actual ET is often constrained by soil water availability and root uptake capacity. As the root-soil interface becomes hydraulically disconnected under drought, transpiration may decline despite strong evaporative demand. Thus, ET is better explained by the coupling of root foraging behavior and soil water retention characteristics in these ecosystems."

**Clarification of Study Scope:**

We appreciate your suggestion regarding the broader physiological purpose of transpiration. While such discussion provides valuable ecological context, the primary focus of our study is on improving remote sensing-based ET algorithm and dataset performance. As such, we may not delve deeply into modeling root water foraging processes. To ensure global applicability, we adopted a simplified yet robust scheme —an essential trade-off in large-scale remote sensing applications.

---

## Author Response (AR1)

**Revision Notes for Manuscript ESSD-2025-137**

We sincerely appreciate all handling editors and referees for their efforts and constructive comments, which have significantly contributed to improving the quality of our manuscript. We have thoroughly revised our manuscript based on the reviewer comments. The detailed point-to-point responses are provided below.

**Response to Reviewer#1's Comments**

*Please refer to the "changes_tracked" version of our manuscript, which is attached by the end of the Notes, to see our detailed edits and revisions.*

*Replies to Comments:*

*1. "a multi-decadal global daily land surface actual evapotranspiration dataset enhanced with explicit soil moisture constraints in remote sensing retrieval"*
    *Could the title possibly be better expressed, thus-?*
    *"Global daily evapotranspiration estimated from land surfaces by remote sensing over multiple decades, including explicit soil moisture constraints to remote data retrieval."*
**Response:**

We sincerely thank the reviewer for the thoughtful suggestion regarding the manuscript title. We agree that the proposed version improves fluency and places helpful emphasis on "global daily evapotranspiration". While our original title was indeed longer, it aimed to reflect the dataset's key characteristics and methodological foundation, consistent with the conventions of Earth System Science Data (ESSD) data description papers. For reference, similar titles in ESSD include:

    • *CAMELE: Collocation-Analyzed Multi-source Ensembled Land Evapotranspiration Data*

    • *A global 5 km monthly potential evapotranspiration dataset (1982–2015) estimated by the Shuttleworth–Wallace model*

    • *A daily and 500 m coupled evapotranspiration and gross primary production product across China during 2000–2020*

    • *A global terrestrial evapotranspiration product based on the three-temperature model with fewer input parameters and no calibration requirement*

In consideration of the reviewer's valuable feedback and to better align with ESSD's style and audience expectations, we have revised the title to: **P-LSHv2: A multi-decadal global daily evapotranspiration dataset enhanced with explicit soil moisture constraints**

We believe this revised title improves clarity and conciseness while preserving the necessary level of detail and methodological specificity. The revised title is now shown in the manuscript (see lines 1-3).

*2. "We integrated this approach into the process-based land surface 20 ET/heat fluxes algorithm (P-LSH, or P-LSHv1), developing an improved version, P-LSHv2. Using observations from 106 global flux towers, we calibrated biome- and climate-specific*

*parameters and quantified moisture constraints across diverse climates and land cover types. P-LSHv2 achieves notable improvements in ET estimation, with a reduced Root Mean Square Error (RMSE) of 0.67 mm d⁻¹ and an increased correlation coefficient (R) of 0.81, outperforming its predecessor, P -LSHv1, particularly in arid regions."*

*A most efficient description of complex processes, but, should (R be R2)?*

**Response:**

We thank the reviewer for the positive feedback and the insightful question. In this context, we used the **Pearson correlation coefficient (R)** to assess the linear agreement between the estimated and observed evapotranspiration (ET) values across flux towers. Since our focus is on evaluating consistency rather than the proportion of explained variance—as would be the case with the coefficient of determination ($R^2$)—we believe that reporting **R = 0.81** is appropriate.

To avoid any potential confusion, we have clarified this explicitly in the revised manuscript. The sentence has been revised as follows (see lines 23-24):

"…P-LSHv2 achieves notable improvements in ET estimation, with a reduced root mean square error (RMSE) of 0.67 mm d⁻¹ and an increased **Pearson correlation coefficient (R)** of 0.81, indicating strong agreement with flux tower observations. As a result of these improvements, P-LSHv2 outperforms its predecessor, P-LSHv1, particularly in arid regions…"

*3. "Leveraging the P-LSHv2 algorithm, we have produced a long-term global daily ET dataset spanning 1982–2023, providing a valuable resource for research on terrestrial water and energy cycles and climate change. The dataset is freely available at https://doi.org/10.11888/Terre.tpdc.301969 (Feng Jin, 2025)."*

*This is a very generous offer of free access to your data, Jin.*

*I just question your choice of the word 'Leveraging'. I know I am a 72-year old, old fashioned bloke who still used printed map books to figure out where to drive in the city, but still have a more than adequate mental map of the whole of SW Australia to call on from long years of driving around helping agriculture. But the point is that the word 'Leveraging' primarily reminds me of the very skilled Aussie tyre fitter who I often had to call upon to change a tyre or two on the government car I was driving around. So maybe for the sake of a broad readership of your extensive paper, the word 'employing' may be a more suitable one in these circumstances? Just a respectful suggestion.*

**Response:**

We sincerely thank the reviewer for the kind words and for the thoughtful suggestion regarding word choice. We appreciate the perspective on the term "leveraging", and agree

that "employing" may read more naturally and be more widely accessible to a broad readership. To improve clarity and tone, we have revised the sentence as follows (see line 27):

"**Employing the P-LSHv2 algorithm**, we have produced a long-term global daily ET dataset spanning 1982–2023…"

We are grateful for the reviewer's attention to both language and accessibility, which contributes meaningfully to improving the manuscript.

*4. "Due to the water potential gradient between leaf and air, water is transported from soil to vegetation roots, and leaves, and then dissipated into the atmosphere through stomata. Therefore, soil water content serves as the direct water pool for vegetation and regulates the magnitude of water extracted by vegetation roots (Feng et al., 2022; Liu et al., 2020b)"*

*This is an eloquent, but oversimplified, physical explanation of evapotranspiration. It requires inclusion of the biological need and purpose of transpiration by plants and the vital role to sensory and growth behaviour that plant root tips play in semi-arid ecologies especially in landscapes with soil types of poor water-holding capacity. This text is extracted from one of the research papers I am developing.*

*"Dexter (1986) described the behaviour of plant roots seeking biopores, some concepts have been put forward, such as "trematotropism" and "oxytropism". Gregory (2009) summarised that 'Roots grow towards areas of higher water potential … and that roots could sense a water potential gradient as small as 0.5 MPa mm–1 so that hydroresponsiveness may contribute to both avoidance of drought stress and modifications to root system architecture'. This knowledge strongly suggest that the soil profile structure needs some degree of heterogeneity varying from loose structure for ease of root exploration to more dense components (clods or ridges) which allow only slow or little root growth and can retain moisture at higher potential; more readily available at times when growing conditions are drier. In a similar manner, roots seek out some nutrients along gradients of their occurrence in the soil profile as nutrients are supplied to the root surface by mass flow and diffusion'."*

*Thus, by logical deduction, the ability of root tips to search out water in the soil profile may have more control on ET that the simple vapour deficit gradient. Additionally, since most of the evaporated water is used to cool leaves on hot afternoons, any undersupply and overheating leads to a breakdown of ET pathways through the plant tissue and a reduction of ET despite a strong VP gradient. Such processes do need to be explained and included in this MS, and there may well be more research of that aspect, since I am not fully up-to date with that research sector.*

**Response:**

We sincerely appreciate the reviewer for the insightful and constructive comments. Your suggestions have significantly enriched our understanding of plant water use strategies and the underlying ecological mechanisms. Below, we summarize the key issues you raised and provide our detailed responses:

(1) Transpiration mechanism oversimplified

The explanation of plant transpiration in the manuscript is overly simplified, focusing mainly on physical processes (i.e., water potential gradients) while neglecting physiological drivers and regulatory mechanisms—especially plant responses under drought constraints.

(2) Active root sensing and hydrotropism

Plant roots are not passive in water uptake but actively sense and grow toward water through physiological mechanisms. The ability to detect subtle water potential gradients and directionally grow plays a key role in maintaining transpiration under moisture-constrained conditions.

(3) Soil structure effects on root uptake

Soil structural heterogeneity substantially influences root architecture and water availability. Loose soils promote root exploration, while compacted structures aid in water retention, thereby shaping effective water uptake and transpiration dynamics.

(4) ET limitation under drought conditions

In dryland ecosystems, root water acquisition may exert a more direct control on ET than atmospheric drivers such as vapor pressure deficit (VPD). Under drought, even high VPD may not lead to higher ET due to limited plant access to water.

In response to these points, we have revised the manuscript as follows:

**Response to Point (1):**

We acknowledge that our previous description of transpiration primarily focused on its physical pathway. In the revised manuscript, we have added further explanation of the physiological regulation of transpiration (see lines 66-68, 72-74). Specifically:

"In addition to this physical gradient, plant transpiration is fundamentally driven by biological needs such as nutrient transport, turgor maintenance, and leaf cooling. These physiological processes are tightly regulated and exert feedback control on stomatal conductance, thereby influencing transpiration dynamics."

"Therefore, soil water content not only serves as the direct water pool for vegetation but also provides the essential hydraulic foundation that supports plant physiological functions. It

ultimately regulates both the capacity and the demand side of transpiration through its dual role in water supply and physiological control."

**Response to Point (2):**

We have revised the description of root water uptake to clarify that roots are not merely passive structures. We have emphasized their sensory and active water-seeking behaviours (see lines 68-72). The revised text includes:

"Importantly, root tips are capable of sensing subtle gradients in water potential (as low as 0.5 MPa mm$^{-1}$), exhibiting behaviors such as hydrotropism to actively seek water in heterogeneous soil profiles(Dexter, 1986; Gregory, 2006). Such sensory responses provide a physiological basis for root foraging behavior, which is particularly important for sustaining transpiration under drought conditions."

**Response to Point (3):**

We agree that soil structural heterogeneity plays a key role in root development and water availability. In our P-LSHv2 algorithm, such heterogeneity is indirectly represented by land cover and climate classifications, which is determined by the parameter $n$. We opted not to use global soil hydraulic properties due to their high uncertainty, but land cover and climate types provide a feasible proxy for large-scale heterogeneity. We have also mentioned the potential of incorporating higher-resolution soil hydraulic properties in future work (see lines 525-534). The following paragraph have been added to the discussion:

"Soil structural heterogeneity plays a crucial role in regulating root development and water availability, which further constrain ET. Looser soils facilitate root penetration, while denser soils can retain water at higher matric potentials, thus extending water availability during dry periods. These physical characteristics influence not only the spatial distribution of roots but also the efficiency of root water uptake and overall transpiration dynamics. Incorporating soil hydraulic properties (e.g., soil water retention curves and hydraulic conductivity) would ideally allow more accurate constraint of soil moisture availability in evapotranspiration estimation. However, global soil hydraulic datasets remain highly uncertain due to limited in situ observations, spatial variability, and differences in pedotransfer function assumptions. To address this challenge, we adopted a simplified quantile-based method to parameterize the soil moisture constraint function in our P-LSHv2

algorithm. Specifically, the parameter n was determined based on land cover and climate types, which serve as proxies for large-scale heterogeneity in both soil properties and vegetation root strategies."

**Response to Point (4):**

We agree that in arid ecosystems, the availability of soil water may limit ET more directly than VPD. This perspective supports our inclusion of explicit soil moisture constraints in the P-LSHv2 algorithm. Our results also indicate that ET in arid ecosystems is highly sensitive to soil water availability. We have added the following explanation in the manuscript (see lines 79-91):

"In arid and semi-arid regions, even under high atmospheric demand (i.e., high VPD), the actual ET is often constrained by soil water availability and root uptake capacity. As the root–soil interface becomes hydraulically disconnected under drought, transpiration may decline despite strong evaporative demand. Although vapor pressure deficit and soil moisture are generally connected through land-atmosphere interactions, their anomalies may be temporally lagged, indicating a partial decoupling. Therefore, relying solely on VPD can misrepresent transpiration dynamics, especially under conditions of prolonged soil dryness or climate extremes. Incorporating explicit soil moisture constraints is essential for improving ET estimation at finer temporal scales, and this need is expected to intensify under global warming scenarios, where soil–atmosphere coupling may become even more unstable."

**Clarification of Study Scope:**

We appreciate your thoughtful comments regarding the broader physiological drivers of transpiration. While such aspects—particularly root and plant functional processes—offer valuable ecological insight, our current study primarily focuses on improving the performance of remote sensing–based ET algorithms and datasets. Given the constraints of global-scale application and data availability, we acknowledge the limitation in fully capturing in-situ root water foraging processes.

We are encouraged by your understanding of these limitations and greatly value your suggestion to promote the application of our estimation system in future studies. In particular, we agree that combining our remote sensing framework with well-designed ground-truthing efforts in environmentally contrasting regions will be essential for further validating and

refining this algorithm. We hope our work can serve as a foundation for such integrative investigations.

**Response to Reviewer#2's Comments**

*Please refer to the "changes_tracked" version of our manuscript, which is attached by the end of the Notes, to see our detailed edits and revisions.*

*Replies to the General Comments:*

*1. This paper describes an update to the established P-LSH ET model, incorporating a new soil moisture constraint and model calibration to improve global performance. Model validation is assessed for global wet and dry climate zones against flux tower measurements and independent watershed level ET estimates to document relative model improvements in relation to the current model (v1) and other global ET records (GLEAM, Penman-Monteith-Leuning). Overall, the results demonstrate clear and meaningful P-LSHv2 ET performance improvement relative to ET observations and the other models. The addition of a model soil moisture constraint more strongly enhances model accuracy in dry climate zones, while providing a more realistic representation of environmental controls on ET trends. The paper is well written and comprehensive, with multiple levels of evidence to support the findings and conclusions, and well illustrated figures and tabular summaries that give the reader a clear understanding of model improvements. I therefore consider the paper to be suitable for publication once the authors address the following the following minor issues.*
**Response:**

We sincerely thank the reviewer for the positive and constructive evaluation. We have carefully addressed all issues raised and provide detailed responses below.

*2. "The authors state that tower measurements, including surface soil moisture, are used to drive and evaluate the ET algorithm (Ln 236-238). However, it's unclear whether the tower level performance assessment (Section 4.2) is based on model ET simulations derived from local tower meteorological measurements or GLDAS inputs. Additional explanation is needed here.*
**Response:**

We appreciate the reviewer's helpful comment. We have clarified the data sources used in Section 4.2 (see lines 378-380). Specifically, in this section related to tower-level performance, the P-LSHv2 model was driven entirely by meteorological measurements from each flux tower, including surface soil moisture, with the only exception being NDVI, which was derived from remote sensing products. This clarification has been explicitly included in the revised manuscript as follows:

"We estimated daily ET at 106 global flux towers using the optimized P-LSHv2 algorithm, driven by tower-based measurements of radiation, meteorology, soil moisture, and remote sensing-based NDVI. The estimated ET was then compared against flux tower measurements for evaluation."

*3. The authors compare model ET performance and soil moisture sensitivity between wet and dry climate zones defined from a global climate aridity index (AI). However, the simple climate AI partitioning groups energy-limited cold land areas, including northern taiga and tundra, into the dry climate category (e.g. Fig. 2) even though these areas have generally wet soils with minimal soil moisture constraints during the short summer growing season. Thus, tundra is grouped with other GRS and OSH dominant land covers even though these other areas may represent much warmer-drier climate zones (e.g. sub-tropical Africa & western CONUS drylands). Failure to distinguish energy limited zones may contribute to the excessive model soil moisture constraint indicated in tundra (Fig. 5) and the corresponding relative ET model underestimation in this region (e.g. Fig. 14). Additional discussion is needed along these lines.*

**Response:**

We sincerely thank the reviewer for the insightful comments on the limitations of the aridity index (AI)-based dry–wet classification, particularly regarding the misclassification of energy-limited ecosystems as "dry" regions, such as the northern taiga and tundra. This concern is well founded. The AI, defined as the ratio of precipitation to potential evapotranspiration (P/PET), is a widely used metric to distinguish between water-limited and energy-limited climates. In high-latitude regions, low temperatures result in low PET, and while annual precipitation may also be low, soils tend to remain moist due to reduced evaporative demand and limited vegetation activity. Consequently, these regions can exhibit low AI values without experiencing true water scarcity. This misclassification may indeed contribute to an overestimation of soil moisture constraints and underestimation of ET in taiga and tundra regions during the growing season. During the short but intense growing season, soils are typically saturated or near saturation, and ET is primarily constrained by energy availability rather than soil water.

While this critique is valid, some caveats should be noted. Despite the presence of wet surface soils in summer, the tundra biome is characterized by an extremely short growing season, low radiation inputs, and permafrost-driven hydrological dynamics that distinguish it from both arid and temperate systems. Therefore, annual ET magnitude and variability in these regions are more strongly influenced by energy constraints and vegetation phenology than by seasonal water availability.

Furthermore, as shown in Fig. 13, both GLEAM and our earlier P-LSHv1 tended to overestimate ET in northern basins. The revised P-LSHv2 algorithm provides more reasonable estimates, despite showing slight underestimation in Fig. 14—an underestimation that is relative to our previous version, not to ground-truth observations.

To ensure global applicability and operational feasibility, the current version of our model uses a parameterization scheme based on vegetation type and climate zone derived

from widely available static datasets. We did not incorporate phonological dynamics or vegetation growth in this version. While this simplification may introduce uncertainty in regions like the taiga and tundra, it represents a necessary trade-off between model complexity and global generalizability. Moreover, evaluating soil moisture constraints at the annual scale, rather than exclusively during the growing season, can still yield meaningful insights for ecosystems with short active periods.

The AI remains an internationally accepted and widely applied metric for global climate classification. Despite its weaker physical interpretation in cold regions, its statistical properties and global consistency make it a practical basis for large-scale comparisons. We recognize that more refined classification systems—potentially incorporating growing season length, vegetation phenology, and permafrost dynamics—could better distinguish energy-limited from water-limited systems. Nonetheless, even hybrid approaches (e.g., combining AI with temperature) may still fall short of fully capturing these complexities.

Given these challenges, we have opted not to modify the AI-based climate zonation in the current version. However, we fully acknowledge the value of more sophisticated climate classifications—particularly in cold regions—and suggest that future efforts explore integrated ecohydrological zoning approaches to better represent energy-limited ecosystems in global ET modeling. The relevant discussion has been added to the revised *Discussion* section to better acknowledge this limitation and guide future work (see lines 576-582). We appreciate the reviewer for highlighting this important issue, which has helped us clarify model limitations and identify future directions for improvement.

*4. Ln 42: Text should be modified similar to: MODIS data do not cover the pre-2000 period and are of insufficient length to represent longer-term interannual variability and trends, and attribution analysis in ET. The revised statement more correctly acknowledges the longer MOD16 ET record available from the NASA Terra satellite. Moreover, while the MODIS record is too short to capture climate "normals" that would require a minimum 30-year span, the data record does represent a comprehensive (500m, 8-day) multi-decadal global operational satellite ET record, which has been used to evaluate more recent interannual variability and trends (e.g. Hall et al. 2023, Roman et al. 2024).*
**Response:**

Thank you for the helpful suggestion. We agree that our previous statement did not adequately assess the time span and utility of the MOD16 ET record. We have revised the sentence to better reflect the MOD16 dataset's temporal coverage and value for assessing recent interannual variability, while also acknowledging its limitations in capturing long-term climate normal. Relevant citations have also been added in our revised manuscript (see lines

42-46).

***Replies to the Specific Comments***
*1. Ln 14: "curcial" should be "crucial".*
**Response:**

Thank you for pointing this out. We have corrected the spelling from "curcial" to "crucial" (see line 14).

*2. Ln 236: Please define what is meant by "surface" soil moisture here; e.g., 0-5cm depth?*
**Response:**

In the case of GLDAS, we used the 0–10 cm soil moisture product. For the flux tower measurements, we selected the shallowest available data from each tower, typically represented by variables such as "SWC_F_MDS_1", "SWC_PI_1", or "SWC_1_1_1", where the "_1" suffix denotes the top soil layer. The measurement depth varies slightly across various towers, but these variables generally represent soil moisture in the uppermost 0–5 cm or 0–10 cm layer.

*3. Ln 497: Include supporting citation on the noted net radiation decline since 2016.*
**Response:**

Thank you for the suggestion. The statement regarding the decline in net radiation since 2016 was based on a preliminary analysis of our input data. However, since we could not identify a well-established peer-reviewed reference to support this trend, we have decided to remove the description to maintain the scientific rigor of the manuscript. The sentence has been revised (see lines 517-519) as follows:

"The trend of P-LSHv2 is comparable to PML (0.68 mm yr$^{-2}$), and higher than GLEAM (0.38 mm yr$^{-2}$). P-LSHv2 ET increased by 0.46 mm yr$^{-2}$ ($p < 0.001$) from 1982 to 2023, although the rate of increase appears to have slowed in recent years."

---

## Referee Report (RR1)

**Review Report of "P-LSHv2: a multi-decadal global daily evapotranspiration dataset enhanced with explicit soil moisture constraints"**

General comments:

This paper provides an improved process-based land surface ET fluxes algorithm integrated with a soil moisture constraint scheme. The parameters are calibrated based on global flux tower observations over various climates and biomes, with uncertainty and sensitivity well quantified. The calibrated model is then used to generate a global daily evapotranspiration dataset, which outperforms its previous version and aligns well with other benchmark products. This new ET dataset will be a valuable reference for global energy budget and water balance studies.

The resubmitted manuscript is well written with high content clarity and comprehensive analysis. The comments from the two reviewers are well addressed, and the modifications in the updated manuscript are proper. Some comments addressed in the discussion, such as the dominant effects between energy vs. moisture constraints in high latitude regions, and the impact of deeper root-zone soil moisture, are interesting and worthy directions for further studies in the future. I recommend that the updated manuscript be accepted for publication.

Specific comments:

1. The type of prior distributions used for the parameters should be mentioned somewhere in the text. It would also be helpful to indicate the prior intervals in Figure 4 to make the comparison between prior and posterior more obvious.

---

## Author Response (AR2)

**Second Round Revision Notes for Manuscript ESSD-2025-137**

**Topic editor comments:** *The revised manuscript presents a well-structured and valuable contribution, with solid methodology and meaningful results. The reviewers raised several minor concerns for the current version primarily related to clarity, figure readability, and transparency of parameter assumptions. These issues are readily addressable and do not affect the overall validity of the work.*

**Response:**

Thank you for the encouraging comment. We sincerely appreciate the editors and reviewers for their constructive feedback. We have thoroughly revised the manuscript in response to all comments. Detailed point-to-point responses are provided below.

**Response to Reviewer#1's Comments**

***Please refer to the "changes_tracked" version of our manuscript, which is attached by the end of the Notes, to see our detailed edits and revisions.***

**Replies to the General Comments:**

*The paper presents an updated ET algorithm and dataset that shows improved performance at flux site, CONUS-basin, and global basin levels. It is a valuable contribution to the literature. I have the following minor comments.*
**Response:**

We sincerely thank the reviewer for the positive evaluation and encouraging feedback. Our point-to-point responses to your comments are listed below.

**Replies to the Specific Comments:**

1. *p11 lines 283-286: NCEP2 is no doubt a well performing dataset in 2015, but the resolution and methodology of global reanalysis has significantly improved since then, and the use of NCEP2 needs additional justification. How may have NCEP2 biases contributed to the biases observed in Figs.11 and 13? Also, the interpolation of NCEP2 from 1.9-degree to 1/12 degree resolution probably resulted in underestimation of the spatial heterogeneity in ET within the original ~1.9degree cells?*
**Response:**

Thank you for your insightful and constructive comments.

**Justification for using NCEP2:** The use of NCEP2 in this study is primarily motivated by its consistency with the dataset used in the original version of our algorithm (Zhang et al., 2015), which facilitates comparability and helps isolate the effects of model improvements from changes in input data. Furthermore, NCEP2 offers a long-term, globally consistent dataset with relatively stable assimilation methods, which suits the multi-decadal scope of this study. We now clarify this rationale in the revised manuscript (see lines 548-551).

**Potential contribution of NCEP2 biases to ET simulation:** We agree that biases in NCEP2 may have contributed to some of the discrepancies shown in the ET spatial patterns. This is likely due to known cold and dry biases in NCEP2, including the underestimation of vapor pressure in certain regions, which could influence energy availability and evaporative demand. We have added a note discussing this in the revised Discussion section (see lines 551-553, 556-560).

**Interpolation and underestimation of spatial heterogeneity:** As you pointed out, the coarse resolution of NCEP2 (1.9°) could indeed limit the representation of fine-scale spatial heterogeneity when interpolated. However, since we mainly used NCEP2 for air temperature, wind speed, and vapor pressure—variables that tend to have relatively smoother spatial gradients—the impact of downscaling on spatial heterogeneity might be less severe compared to variables like soil moisture. Nonetheless, we acknowledge this as a limitation in the Discussion (see lines 553-556).

To address these limitations, we have added a paragraph in the Discussion section acknowledging the potential influence of NCEP2 biases on ET simulation and spatial patterns, and expressing our plan to incorporate higher-resolution and more modern reanalysis datasets (e.g., ERA5 or MERRA-2) in future work.

2. *The Feng et al. (2022) P-LSH algorithm has a radiation multiplier but no daylight temperature multiplier. Could the author explain why the swap in this paper? Also, I am not sure what is the meaning of daylight temperature - is it daytime temperature?*
**Response:**

Thank you for your valuable comment. Regarding the use of stress factors, both our previous work (Feng et al., 2022) and this study include daylight temperature ($T_{day}$) as stress factor—by which we indeed mean daytime temperature. In Feng et al. (2022), we incorporated five stress factors in the Jarvis-Stewart-based stomatal conductance model: $T_{day}$, VPD, $CO_2$, solar radiation ($R_s$), and soil moisture (SM). In this study, we excluded the radiation scalar and retained $T_{day}$, VPD, $CO_2$, and SM.

This change was primarily motivated by following considerations. First, our preliminary analysis showed that $T_{day}$, VPD and SM exert the strongest influence on stomatal conductance across most vegetation types and climatic zones, while radiation exhibited limited marginal influence in Jarvis-Stewart model. Excluding radiation improved model parsimony without significantly compromising performance.

Second, at the global scale, incorporating fewer but more robust input variables enhances model stability and reduces the propagation of uncertainties from remote sensing data. In many cases, radiation is strongly correlated with other meteorological variables (e.g., temperature, VPD), and its effects can be partially captured indirectly.

Finally, it is worth noting that radiation is already explicitly accounted for in the Penman-Monteith framework used for ET computation. Therefore, its exclusion from the Jarvis-Stewart stomatal conductance formulation does not mean it is neglected in the overall ET estimation framework.

Based on the above analysis, we eventually eliminated the radiation scalar in this study.

3. *The flux tower dots are overlapping and hard to read in Fig. 7. Also the caption does not explain what the "classified statistical values" are in the right panels. It is suggested to re-draw all the data as boxplots.*
**Response:**

Thank you for the valuable suggestion. We initially chose dot plots to display the individual statistics of each flux towers, aiming for a more intuitive understanding of data distribution. However, we acknowledge that overlapping markers reduce readability. To address this, we have reduced the marker size and applied transparent face colours in the revised figure to enhance clarity (see line 387).

Regarding the use of boxplots, we note that some subcategories contain fewer than five data points, which would make boxplots statistically unreliable and potentially misleading. In such cases, boxplots may not meaningfully represent quartiles or potential outliers, which could lead to misinterpretation.

Regarding the term "classified statistical values" in the right panels, it refers to statistical metrics summarized across all towers, dry towers, and wet towers, respectively, allowing for a clearer comparison of model performance across different zones. We have revised the caption to clarify this definition (see lines 388-391).

4. *Fig. 8: It is virtually impossible to see the P-LSHv2 line due to color choice. Please use a more visible color.*
**Response:**

Thank you for your suggestion. We have revised Figure 8 to improve the visibility of the P-LSHv2 line by using a more distinguishable and clearer colour (see line 406). We believe this change enhances the figure's readability and better highlights the performance of our algorithm.

5. *Fig. 9: all the symbols are overlapping. Perhaps shrinking the marker size and setting the face color to transparent will make the reader better distinguish the blue, pink, and dark brown symbols.*
**Response:**

Thank you for the helpful suggestion. In the revised version of Figure 9, we have reduced the marker size and set the marker face colour to transparent as recommended. These adjustments improve the visual clarity and make it easier to distinguish between the blue, pink, and dark brown symbols. Please refer to the revised Figure 9 (see line 422).

**Response to Reviewer#2's Comments**

***Please refer to the "changes_tracked" version of our manuscript, which is attached by the end of the Notes, to see our detailed edits and revisions.***

**Replies to the General Comments:**

*This paper provides an improved process-based land surface ET fluxes algorithm integrated with a soil moisture constraint scheme. The parameters are calibrated based on global flux tower observations over various climates and biomes, with uncertainty and sensitivity well quantified. The calibrated model is then used to generate a global daily evapotranspiration dataset, which outperforms its previous version and aligns well with other benchmark products. This new ET dataset will be a valuable reference for global energy budget and water balance studies.*

*The resubmitted manuscript is well written with high content clarity and comprehensive analysis. The comments from the two reviewers are well addressed, and the modifications in the updated manuscript are proper. Some comments addressed in the discussion, such as the dominant effects between energy vs. moisture constraints in high latitude regions, and the impact of deeper rootzone soil moisture, are interesting and worthy directions for further studies in the future. I recommend that the updated manuscript be accepted for publication.*

**Response:**

We sincerely thank the reviewer for the positive and encouraging comments on our revised manuscript. We are grateful for the recognition of the improvements made in the algorithm, data calibration, and analysis. Detailed point-to-point responses are provided below.

**Replies to the Specific Comments:**

1. *"The type of prior distributions used for the parameters should be mentioned somewhere in the text. It would also be helpful to indicate the prior intervals in Figure 4 to make the comparison between prior and posterior more obvious.*

**Response:**

Thank you for the valuable comment. Since no prior information was available for the parameters, we assumed uniform distribution as prior distribution, which was subsequently updated to posterior distributions through the Bayesian-based methods. This clarification has been explicitly included in the revised manuscript (see lines 212-213).

Additionally, we revised Figure 4 to include dashed black lines indicating the lower and upper bounds of the prior intervals (see lines 334-337), to facilitate a clearer comparison between the prior and posterior distributions.